**METHOD**

# Regulatory analysis of single cell multiome gene expression and chromatin accessibility data with scREG

Zhana Duren[1*†] , Fengge Chang[1†], Fnu Naqing[1], Jingxue Xin[2], Qiao Liu[2] and Wing Hung Wong[2*]

†Zhana Duren and Fengge Chang contributed equally to this work.

*Correspondence:
zduren@clemson.edu;
whwong@stanford.edu

[1] Center for Human Genetics and Department of Genetics and Biochemistry, Clemson University, Greenwood, SC 29646, USA
[2] Department of Statistics, Department of Biomedical Data Science and Bio-X Program, Stanford University, Stanford, CA 94305, USA

## Abstract

Technological development has enabled the profiling of gene expression and chromatin accessibility from the same cell. We develop scREG, a dimension reduction methodology, based on the concept of *cis*-regulatory potential, for single cell multiome data. This concept is further used for the construction of subpopulation-specific *cis*-regulatory networks. The capability of inferring useful regulatory network is demonstrated by the two-fold increment on network inference accuracy compared to the Pearson correlation-based method and the 27-fold enrichment of GWAS variants for inflammatory bowel disease in the *cis*-regulatory elements. The R package scREG provides comprehensive functions for single cell multiome data analysis.

**Keywords:** Single cell multiome, *cis*-Regulatory potential, Dimension reduction, cis-Regulatory networks

## Background

Recent advances in technology enable one to study heterogeneous mixtures of cell populations at the single cell level. Single cell RNA sequencing (scRNA-seq) [1] provides whole genome transcription profiling and single cell ATAC-seq (scATAC-seq) [2] identifies accessible chromatin regions at the single cell level. Integrative analysis of scRNA-seq with scATAC-seq could identify the subpopulations more accurately and provides more detail about the gene regulation [3–7]. Traditionally, expression profiling and accessibility profiling are done separately on different sub-samples from the heterogeneous population. To jointly analyze these two types of data, all of these methods require a linking function between cis-regulatory elements (REs) and target genes (TGs). The linking functions were either based on genomic distance or external data. For example, SOMatic [5] links the RE to the nearest gene; the RE-TG connection in MAESTRO [6] is defined as an exponentially decreasing function of their distances; many other methods including Seurat [8] use the gene activity score defined in Cicero [9]; and our previously

developed methods Coupled NMF [3] and DC3 [4] learn the RE-TG connection from external bulk data from diverse cellular contexts and bulk 3D chromatin contact data, respectively. Even though many methods have been proposed, our ability to infer cis-regulation is fundamentally limited by the fact that these two types of genomics features are not measured on the same cell. Fortunately, recent technological development allows joint profiling of gene expression and chromatin accessibility on the same cell [10–14]. We believe more and more such types of data will be generated as the Chromium platform (10X Genomics) provides kits and protocols to make it more convenient to use (sc-multiome) [15].

In the past few years, many fancy and powerful methods have been developed for clustering, 2-dimension embedding, and trajectory/pseudotime analysis of single cell genomics data such as matrix-based algorithms [3, 4, 16], graph-based algorithms [8, 17, 18], probability-based methods [9, 19, 20], and neural network-based methods [5, 7, 21–24]. Several methods are also designed for comparative analysis across multiple conditions [25–28] and time course experiments after stimuli [29, 30]. Most of those methods take as input a lower-dimensional representation of cells which is typically obtained in an initial dimension reduction step. For example, many scRNA-seq analysis methods use principal component analysis (PCA) for dimension reduction [31], and some scATAC-seq analyses use latent semantic analysis (LSA) [32] for dimension reduction. In this paper, we address the question of how to perform dimension reduction on single cell multiome data where simultaneous expression and accessibility data is measured on each cell. The simplest approach is to apply standard dimension reduction methods on two data types to produce the corresponding reduced-dimension feature vectors, and then concatenate the two feature vectors to obtain the final feature vector for each cell. However, this simple approach does not leverage the key advantage of sc-multiome data, namely that the activity of RE and TG are measured in the same cell, so that it provides the cis-regulatory information in each cell. Preserving the *cis*-regulatory information should be an important requirement of the dimension reduction step, but this important issue is not taken into consideration in previous methods [16, 18, 33, 34].

The main contribution of this paper is to fill this gap by the introduction of a matrix factorization-based approach designed to preserve regulatory information at the single cell level. Specifically, besides the within-modality information, we also use the cross-modalities information in the dimension reduction by introducing a new concept of *cis*-regulatory potential (CRP) for each pair of regulatory elements and target genes. Here, the CRP is defined as the sum of the accessibility of regulatory element (RE) and expression of target gene (TG) multiplied by the weight dependent on their genomics distance. The advantage of including the concept of CRP is that it helps to capture the *cis*-regulatory information neither included in gene expression alone nor included in the chromatin accessibility alone. In addition, the CRP has much fewer drop-outs so it helps to denoise. Based on the CRP concept, we designed a non-negative matrix factorization (NMF)-based optimization model to project the cells into a lower dimension space. Several NMF-based methods [3, 4, 35, 36] have been developed for dimension reduction and clustering of single cell genomics data and have shown great advantages in data integration. In addition to the gene expression matrix and chromatin accessibility matrix, we also include the *cis*-regulatory potential matrix as one input of the optimization model

to use a lower dimension to represent a cell. After this dimension reduction, downstream analyses such as clustering and 2-dimensional embedding of cells can be performed based on the reduced-dimension features. In addition to dimension reduction, the *second contribution* in this paper is the inference of the subpopulation-specific *cis*-regulatory networks. Based on the concept of CRP, we achieve much higher inference accuracy of RE-TG interaction than the Pearson correlation coefficient (PCC)-based method. A further *contribution* of this work is that the developed methods are implemented into a comprehensive toolkit scREG for the analysis of single cell multiome gene expression and chromatin accessibility data. Our R package scREG enables end-to-end analysis of single cell multiome data, including the functionality of dimension reduction, cell clustering, 2-dimension embedding, regulatory network inference, and interactive visualization. Finally, we use the subpopulation-specific regulatory networks to interpret the disease-associated loci of inflammatory bowel disease (IBD) to demonstrate the usefulness of the scREG inferred regulatory networks.

## Results

### scREG: a computational method for single cell regulatory analysis from multiome data

We propose a computational method for integrative analysis of single cell multiome gene expression and chromatin accessibility data. Figure 1 shows the schematic overview of the scREG analysis workflow. Our software scREG takes as input the read count matrices of gene expression and chromatin accessibility measured on the same cells which are the same format as the standard output of 10X genomics CellRanger software. The output of scREG is a lower-dimensional representation of cells, clustering label, 2-D embedding, and subpopulation-specific *cis*-regulatory networks. The R package also has an interactive visualization function to plot the peaks, genes, and *cis*-interaction in a subpopulation-specific manner for a given genomics range. scREG processes the input data in three main steps: joint dimension reduction, cell clustering, and regulatory network inference.

First, we develop a non-negative matrix factorization (NMF)-based optimization model to reduce the dimension of multiome data with $m_1$ genes and $m_2$ peaks to a common K dimension matrix (default value of K is 100). The gene expression data E matrix and chromatin accessibility data O matrix could be treated as two different modalities and thus those need to be integrated. To capture the cross-modalities information, we define a cell level index *cis*-regulatory potential (noted as *R* matrix, rows represent peak-gene pairs and columns represent cells) to measure the regulatory strength of a peak to a TG in a cell. We project the three different data matrices E, O, and R to a common reduced dimension space by an NMF-based optimization model (see the Joint dimension reduction" section). Specifically, we factorize the three data matrices into products of modality-specific factor profiles ($W_1$, $W_2$, and $W_3$ for E, O, and R respectively) and a common low dimension representation of the cells (the H matrix).

Second, we cluster the cells based on the reduced dimension matrix. Specifically, we calculate the similarity between cells based on the reduced dimension H matrix and construct a k-nearest neighbor graph. To detect rare populations, we transfer this graph to a weighted graph where the weight between two nodes is defined as the Jaccard similarity of their neighbors in the k-nearest neighbor graph. The Louvain algorithm [37]

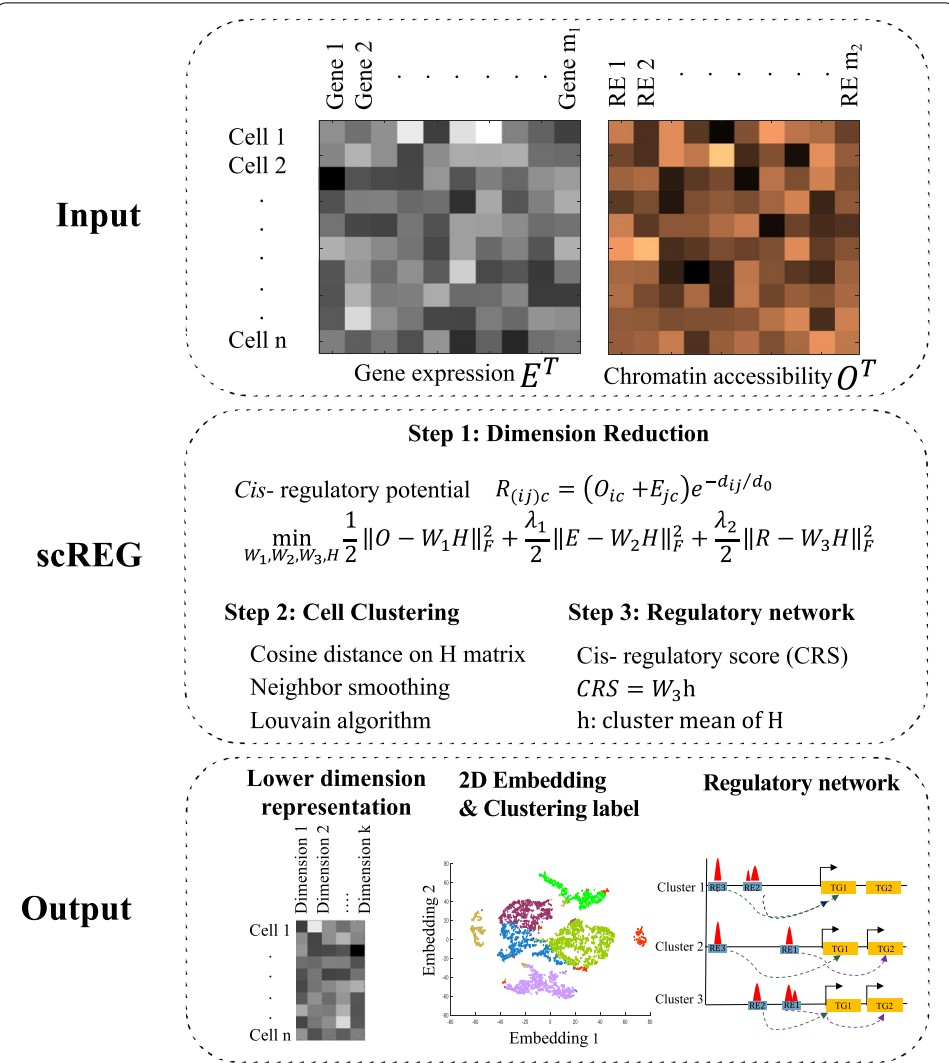

**Fig. 1** Schematic overview of the scREG. The scREG takes matrices of gene expression and chromatin accessibility (E and O respectively) measured on the same cells as inputs and process the multiome data by three steps: dimension reduction, cell clustering, and regulatory network inference. First, a cell level index cis- regulatory potential matrix R is defined, indicating the regulatory strength of a peak to a target gene. Rows of R matrix represent preselected peak-gene pairs and columns represent cells. Here, the index *i* represent the *i*th peak, *j* represents a gene, and *c* represents a cell. Then, three matrices (E, O, and R) are factorized into products of domain specific profiles (W1, W2, and W3 for E, O, and R respectively) and a common low dimension representation of the cells (the H matrix) by NMF-based optimization model. Based on the reduced dimension matrix, we do cell clustering by Louvain algorithm and visualize the cells into 2D space by Umap. For each peak-gene pair, a *cis*- regulatory score (CRS) was defined by average of the cis- regulatory potential over cells from the same cluster. Subpopulation specific *cis*- regulatory networks were identified based on the CRS scores.

was applied on this weighted graph to identify the cell populations (see the "Clustering of cells" sections).

Third, we define a *cis*-regulatory score (CRS) for each peak-gene pair in each cluster by an average of the *cis*-regulatory potential over cells from the same cluster (see the "Reconstruction of the *cis*-regulatory networks" section). We select the top 10,000 peak-gene pairs in each subpopulation to construct the *cis*-regulatory network and identify

regulatory elements (REs) as peaks regulating at least one gene in a given subpopulation. To obtain more accurate regulatory information, we merge cells from the same subpopulation and perform peak calling on each subpopulation. We replace the REs with cluster-specific peaks, which are shorter and more accurate than the original REs, to obtain the final subpopulation-specific *cis*-regulatory networks.

### scREG performs cross-modalities dimension reduction by data integration

To assess whether the scREG method can reduce dimensions efficiently, we apply our method to peripheral blood mononuclear cells (PBMC) multiome data from 10X genomics (see the "PBMC 10K data" section). We use the joint NMF-based optimization model in scREG to reduce the dimension of data into 100 dimensions. To validate the results, we used the cell-type labels that were annotated by the 10X Genomics R&D team as surrogates for ground truths [34] and calculated silhouette index (SI) for each cell based on the reduced dimension matrix. A higher SI value indicates that the cell is more similar to cells sharing its label than those not sharing its label. First, we compare our method with dimension reduction methods based on a single dataset PCA on scRNA-seq, and LSA on scATAC-seq. As the dataset contains 14 cell types, we choose the top 14 dimensions of PCA and LSA. To make the results comparable, we perform PCA on our reduced dimension matrix, which is 100 dimensions, to reduce it to the same dimension as the other two methods. Figure 2A shows that scREG achieves higher SI than PCA RNA, and LSA ATAC on 83.39%, and 83.37% of the cells, respectively.

The increment in SI is cell type-dependent. Figure 2C and Additional file 1: Fig. S1 show the distribution of SI of different methods on different cell types. For CD56 bright NK cells, naive B cells, naïve CD4 T cells, and plasmacytoid DC, the SIs are increased in scREG compared to all the other methods (Fig. 2B). The scREG achieved the best performance on 9 out of 14 cell types. Additional file 1: Fig. S1 shows the comparison of the average SI on all the cell types by different dimension reduction methods, except for the effector CD8 cell, on which all methods generally have poor performance. Our method scREG performs better on most of the cell types, with SI range from 0.2371 to 0.9758, and achieves the highest average SI score across cell types (0.5614).

One alternative way to perform dimension reduction is to perform dimension reduction separately on each type of data first and concatenate them together. We construct a new lower dimension representation by concatenating 7 dimensions of PCA on scRNA-seq and 7 dimensions of LSA on scATAC-seq and compare this with scREG. These newly constructed 14 features perform lightly better than the two methods that perform dimension reduction separately but are inferior to scREG on 83.14% of the cells (Fig. 2A).

To evaluate the importance of the *cis*-regulatory potential, we compare our method with three other methods that do not use cis-regulatory information. First, we build a baseline model of scREG by removing the *cis*-regulatory potential term (the 3rd term in eq. (1) of Methods) from the optimization model and then perform the same optimization and dimension reduction as before. The other two methods for comparison are scAI [16] and MOFA+ [33]. We use these three baseline methods to reduce the dimension to 100 and perform PCA to further reduce it to 14 dimensions. The result shows that including the *cis*-regulatory term improves performance: scREG increases SI on 72.08%,

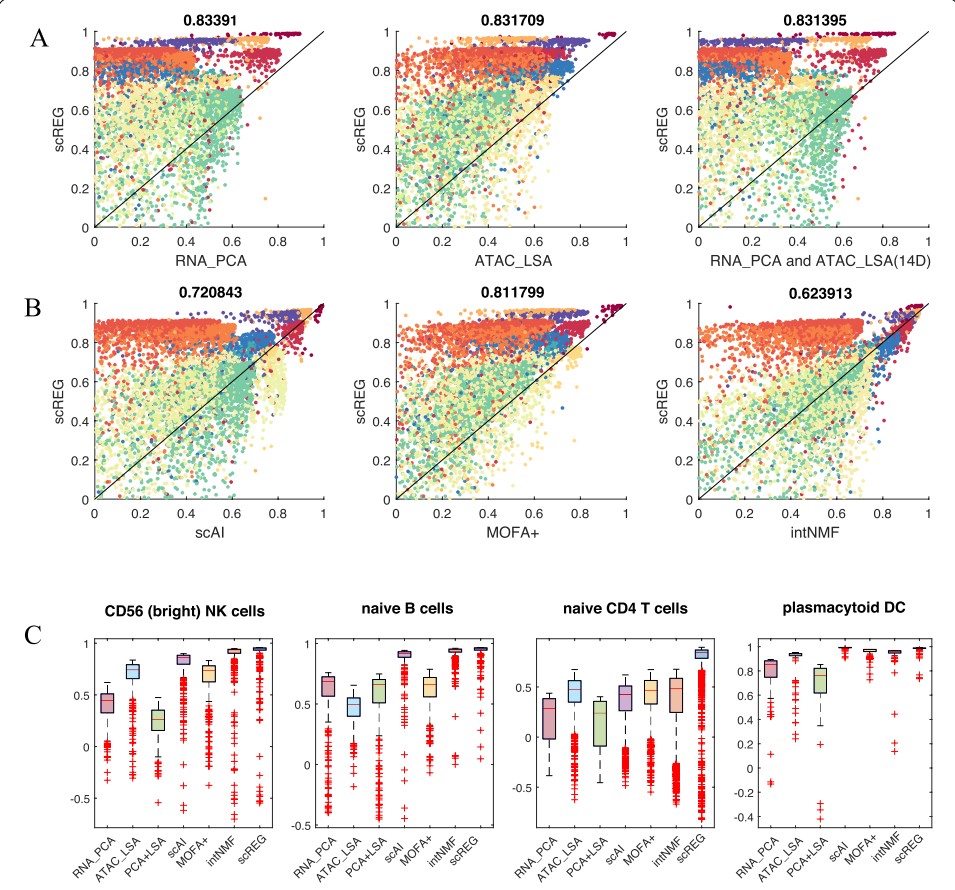

**Fig. 2** Benchmarking scREG with other dimension reduction Methods. **A**, **B** Scatter plots visualize the silhouette index (SI) of following methods versus scREG: PCA for RNA data, LSA for ATAC data, concatenation of top PCs of RNA and top factors of ATAC, scAI, MOFA+, and intNMF, which differ from scREG by lacking the cis-regulatory potential term. Each dot represents the SI of one cell. Each scatter plot compare scREG with another dimension reduction method. *Y* axis represent the SI from scREG and *X*-axes represent that from alternative methods. SI represents the similarity of each cell with the cells in same cell type compared to those from other cell types. The percentage of cells with higher SI in scREG is labeled top of the figures. Color of dots represent true label of cells as shown in Fig. 3B. **C** Comparison of SI values from 7 different methods in some cell types. cells achieve higher SI values when using scREG

81.18%, and 62.39% of cells compared to scAI, MOFA+, and scREG baseline, respectively (Fig. 2B). Figure 2B, C and Additional file 1: Fig. S1 show that the baseline methods without cis-regulatory potential are inferior to the scREG but perform better than the other methods that take a single dataset as inputs. This result suggests that we achieve more accurate low dimension representation by integrating both gene expression and chromatin accessibility data, and the accuracy would be further improved if we include the *cis*-regulatory potentials into the formulation of the NMF-based optimization model.

## scREG identifies the cell populations with high accuracy

To evaluate the performance of the cell clustering aspect of our method, we compare the scREG with Cell Ranger ARC V1.0, Seurat V4.0, scAI, MOFA+, and the baseline scREG method. Cell Ranger ARC analyzes the gene expression and chromatin accessibility separately and outputs two clustering labels, while scREG scAI, MOFA+, and

Seurat perform a joint analysis of the two types of data and output one clustering label. Figure 3A and Additional file 1: Fig. S2A show the UMAP and inferred clustering labels from each of the five methods. Figure 3B and Additional file 1: Fig. S2B show the corresponding UMAPs colored by the surrogate ground truth labels. From the clustering and Umap visualization, we see the scREG gives a consistent label with the surrogate ground truth while alternative methods have obvious misclassification. The naive CD4 T cells and the naive CD8 T cells are not separated in Cell Ranger RNA-seq clustering but separated in Cell Ranger ATAC-seq clustering and the joint clustering methods scREG and Seurat. The non-classical monocytes and the myeloid DC are not separated in both RNA-seq and ATAC-seq clustering but separated in the three joint clustering

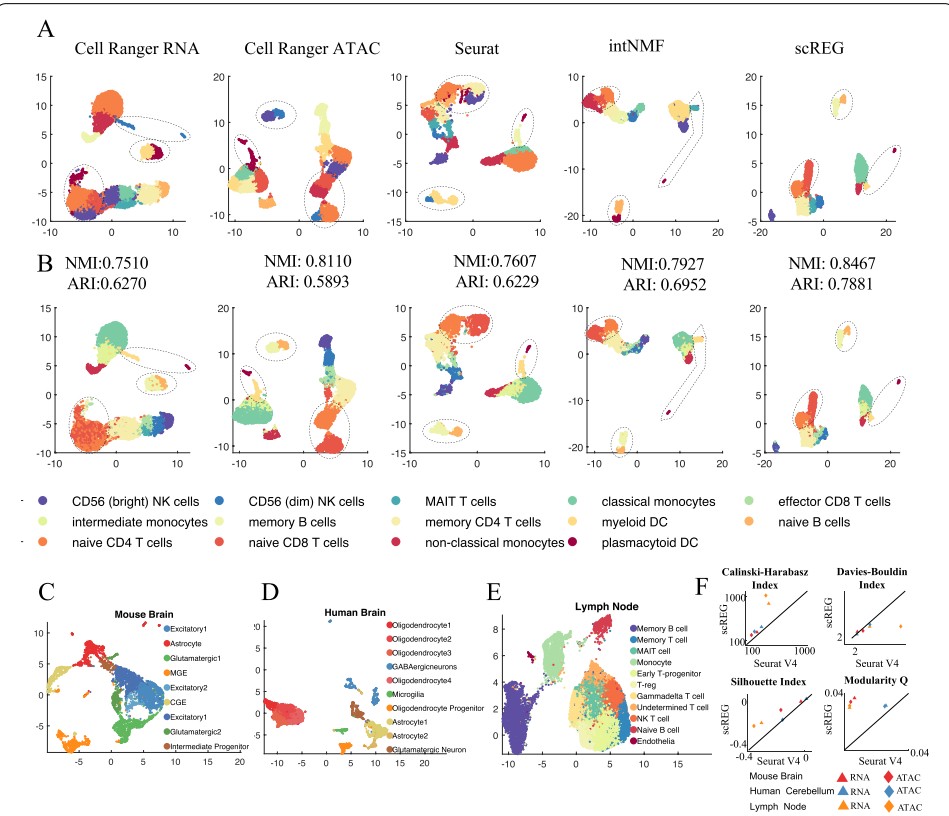

**Fig. 3** Evaluation of the performance of cell clustering. **A** Scatter plot visualize the Umap embedding colored by clustering label from different methods including Cell Ranger on gene expression, Cell Ranger on chromatin accessibility, Seurat V4, intNMF, and scREG. **B** Same Umap as shown in **A** but colored by the surrogate ground truth. We see Cell Ranger RNA-seq did not distinguish naive CD4 T cells from the naive CD8 T cells, and CD56 (dim) NK cells from the effector CD8 T cells. ATAC-seq failed to separate non-classical monocytes and the myeloid DC, while scREG separates them clearly. In Seurat, the boundary between memory B cells and naïve B cells is shifted so a large proportion of memory B cells are labeled as naïve B cells. Clustering performance also assessed by calculating normalized mutual information (NMI) and adjusted Rand index (ARI) based on the surrogate ground truth. **C**–**E** scREG clusterings on 10X multiome data from human cerebellum, mouse E18 brain, and lymph node from B cell lymphoma. The clustering results are consistent with the known cell types and marker genes' expression. **F** The comparison of scREG with Seurat by four different clustering evaluation metrics on three datasets. The distance among cells are calculated as Euclidean distance on the top 20 principal components of gene expression and chromatin accessibility, respectively. *X* axis represents the metric calculated based on Seurat clustering label, and Y axis represent that from scREG clustering. Colors represent different data sets and shape represents different data type (triangle for scRNA-seq and diamond for scATAC-seq). A lower Davies-Bouldin index indicate better clustering, but the other three metrics are the higher the better. The scREG perform better for all case than the Seurat

methods. Memory B cells and naïve B cells are separated clearly in ATAC-seq clustering, but the boundary is not clear in RNA-seq clustering. In Seurat, the boundary between memory B cells and naïve B cells is shifted so a large proportion of memory B cells are labeled as naïve B cells. In scAI clustering, there is a subpopulation that is a mixture of naive CD4 T cells, naive CD8 T cells, and memory CD4 T cells. Memory B cells and naïve B cells are not separated in MOFA+ clustering. These two cell types are separated clearly in scREG_baseline and scREG. To evaluate the clustering results systematically, we calculate the normalized mutual information (NMI) and adjusted Rand index (ARI) based on the surrogate ground truth. It is seen that scREG achieves the highest NMI and ARI compared to other methods. It is worth to notice that all the clustering methods are compared under their default resolution parameter setting, which is 0.8 for Seurat and 1 for all other methods. As clustering accuracy is highly affected by the resolution provided to Louvain clustering, we also compared the clustering performance of these five methods under different resolutions ranging from 0.2 to 2.0 (Additional file 1: Fig. S3). The scREG performs very robust under different resolution parameters and achieve the highest performance among all the method we compared.

We also use scREG and alternative methods on 10X multiome data from the human cerebellum, mouse E18 brain, and lymph node from B cell lymphoma (Fig. 3C–E). The clusters from scREG are consistent with the known marker genes' expression (Additional file 1: Fig. S4-S6). As ground truth labels are not available for these data, we use an adjusted internal clustering evaluation to compare the clustering of scREG with Seurat. It is not fair to compare scREG clustering with Seurat clustering by calculating internal clustering evaluation metrics. The reason is that, when ground truth is not available, the internal clustering evaluation metrics calculated on different distance matrices are not comparable. This is an underappreciated statistical issue, so we generate two artificial examples to illustrate this (please see Additional file 2 for more detail). From example 1, we see the silhouette index is increased but the real clustering accuracy is decreased. This is because internal clustering evaluation metrics are designed for the comparison of different clustering methods (or different parameters) based on the same cell-cell distance matrix. Since internal clustering evaluation has to be performed on the same distance matrix, we have to choose a common embedding space to compare two clusterings. Here, we use the top 20 principal components (PCs) of scRNA-seq and scATAC-seq to calculate the cell-cell distance. For a given clustering, we can say it is a good clustering if cells with the same labels are close to each other in both modalities; we say it is a bad clustering if cells with the same labels are not close to each other in any modalities. Example 2 in the Additional file 2 has illustrated this. It is possible that method A is better than method B on one modality but worse in the other modality. In this case, we cannot evaluate these two methods. For each clustering method, we compute four different clustering evaluation metrics based on top 20 PCs of scRNA-seq and scATAC-seq on the three datasets (Additional file 1: Fig. S7-S9). Figure 3F shows the comparison of scREG with Seurat. It shows scREG performs better than Seurat for most cases.

We also tested our method on the bone marrow mononuclear cells data from the NeurIPS competition [38]. which include 22,463 cells with known cell labels. Additional file 1: Fig. S10 shows the clustering and 2D embedding results and comparison with Seurat. The clustering of scREG is more consistent with the ground truth label and

achieves 0.7649 in NMI and 0.6549 in ARI, which are higher than Seurat (NMI = 0.7409, ARI = 0.5419). Overall, scREG identifies cell populations with high accuracy on different datasets.

### scREG constructs subpopulation specific cis-regulatory networks

To assess the *cis*-regulatory network inference of our method, we evaluate the predictions in several cell populations where experimental *cis*-regulation is available. First, we download variant-gene links defined by the expression quantitative trait loci (eQTL) of CD14 positive monocyte cells [39] and use them to validate the RE-TG prediction of classical monocyte clusters. For each peak-gene pair in the *cis*-regulatory potential $R$ matrix, we have a predicted *cis*-regulatory score in each cluster. Taking the eQTL data as ground truth, we plot the receiver operating characteristic (ROC) curve and precision-recall (PR) curve by sliding the *cis*-regulatory score (Fig. 4A, B). As a baseline method for comparison, we calculate the Pearson correlation coefficient (PCC) between the expression of a gene and the accessibility of peak within 1 Mb of the gene's transcriptional start site. Our method achieves 0.81 area under the ROC (AUROC) curve and 0.46 area under the PR (AUPR) curve, while the AUROC and AUPR of the baseline method are 0.56 and

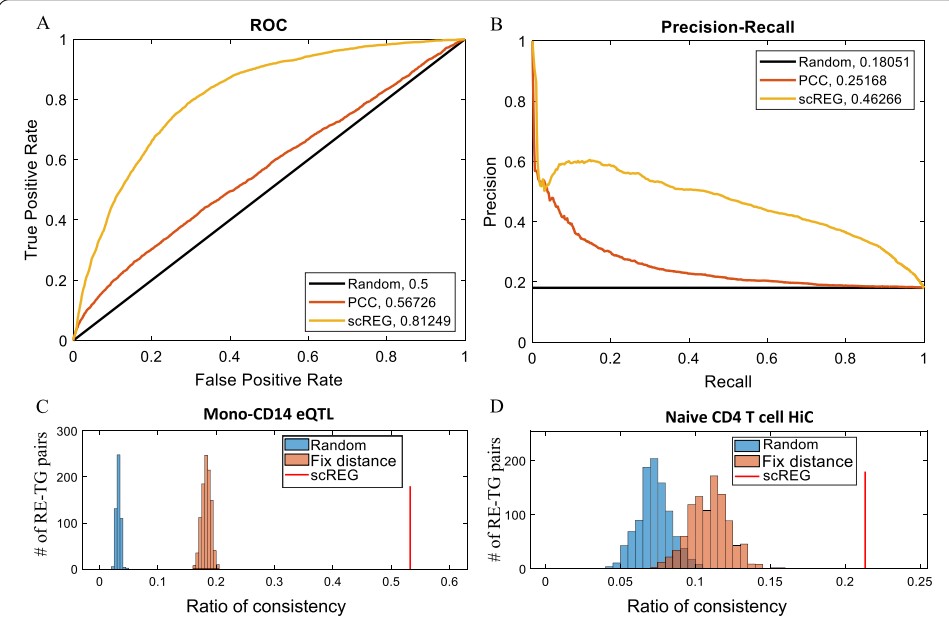

**Fig. 4** Validate the RE-TG prediction. **A**, **B** Receiver operating characteristic (ROC) curve and precision-recall (PR) curve by taking the eQTL data of CD14 positive monocyte cells as ground truth to validate the prediction of scREG. Curves were plotted by sliding the predicted cis- regulatory score of 100,000 peak-gene pairs. As an alternative method, the Pearson correlation coefficient (PCC) between the expression of a gene and the accessibility of peaks within 1 Mb of the gene's transcriptional start site was calculated. We can see scREG prediction achieves 0.81 area under the ROC (AUROC) curve and 0.46 area under the PR (AUPR) curve, which are much higher than that from PCC (0.56 and 0.25 respectively). **C** Comparison of the precision of scREG with random selections. Ratio of consistency (precision) was the percentage of RE-TG pairs validated by eQTL for those REs that linked to at least one gene in eQTL. Red line represents the ratio of consistency of scREG, the blue distribution represents that from 1000 random selection of peak-gene pairs in 1 Mb distance, and the orange distribution represents that from 1000 random selection of peak-gene pairs but restricting the distribution of distance between peak to genes is the same as the scREG predictions. **D** Validation of RE-TG prediction by HiC data on Naïve CD4 T cell. All metrics are calculated same as in **C** but replacing the eQTL by HiC

0.25, respectively. Our method identified 9067 REs in classical monocyte subpopulations, and 1998 of them overlapped with eQTL variants. For those REs that overlapped with eQTL variants, 53.25% of our predictions are connecting them to the same genes as eQTL (Fig. 4C). If we randomly select genes from a 1 Mb distance, this percentage would be 3.26% (16.33-fold), and it would be increased to 18.31% (2.91-fold) if we force the selected peak-gene to have the same distance distribution with our predictions.

Next, we use promoter capture HiC data to validate our predictions. We downloaded the promoter capture HiC data of 7 primary blood cell types [40] and compare them with the *cis*-regulatory networks for the corresponding cell types that we inferred from the PBMC data. Figure 4D and Additional file 1: Fig. S11 shows the precision of our method is 2-fold higher than the randomly selected peak-genes pairs which have the same distance distribution as our predictions. Additional file 1: Fig. S12 shows our method achieves much higher AUROC and AUPR than PCC. The strong performance in the peak-gene link suggests that scREG is effective in inferring regulatory relations.

The *cis*-regulatory networks are highly cell-type-specific. Additional file 1: Fig. S13 shows the Jaccard similarity of clusters in terms of cis-regulation. The average Jaccard similarity between clusters is 0.4760. Hierarchical clustering analysis shows similar cell types have a similar cis-regulatory network. For example, the average similarity between four T cell clusters is 0.7783, and group to one cluster; the similarity of two B cell subpopulations is 0.79; the similarity of two NK cell subpopulations is 0.79; similarity of two monocytes subpopulation is 0.64.

## scREG shed light on the interpretation of disease-associated loci

The *cis*-regulatory network inferred from single cell multiome data may provide new insights for the interpretation of disease-associated loci. To demonstrate this, we download 376 fine mapped variants with a posterior inclusion probability greater than 0.1 for inflammatory bowel disease (IBD) [41, 42]. These fine-mapped GWAS variants showed high enrichment (odds ratio in the range of 9.56 to 27.45) in REs from subpopulations of PBMC data (Fig. 5A, see the "Enrichment of GWAS variants" section) [43]. As a baseline for comparison, we use all peaks from each subpopulation to do the same analysis. As a result, the enrichment odd ratios from all peaks are 1.14-3.02 fold (on average 1.82-fold) lower than the peaks from our networks. These results show that the context specific regulatory network from scREG could improve the interpretation of the disease-associated loci.

The *cis*-regulatory networks produced by scREG connect 34 fine-mapping variants to 32 genes in 12 cellular contexts. These 32 genes include three transcription factors IRF4, IRF8, and CEBPB. Variants rs913678, rs4811031, and rs6063502 are linked to CEBPB in non-classical monocytes; variant rs6935510 in chr6 is linked to IRF4, and variant rs11640143 in chr16 is linked to IRF8 in plasmacytoid dendritic cells. Our scREG package has an interactive visualization function that takes the genomics region as input and plots the genes, peaks, and interactions in the given range. It includes the genes, the raw peaks from all cells (before clustering), peaks of each cluster from MACS2, and the predicted peak-gene association in each cluster. Figure 5B shows the track of around the variant rs11640143 and IRF8. The variant rs11640143 is in an accessible region in myeloid DC, memory B cell, naïve B cell, and plasmacytoid DC cells (highlighted blue in

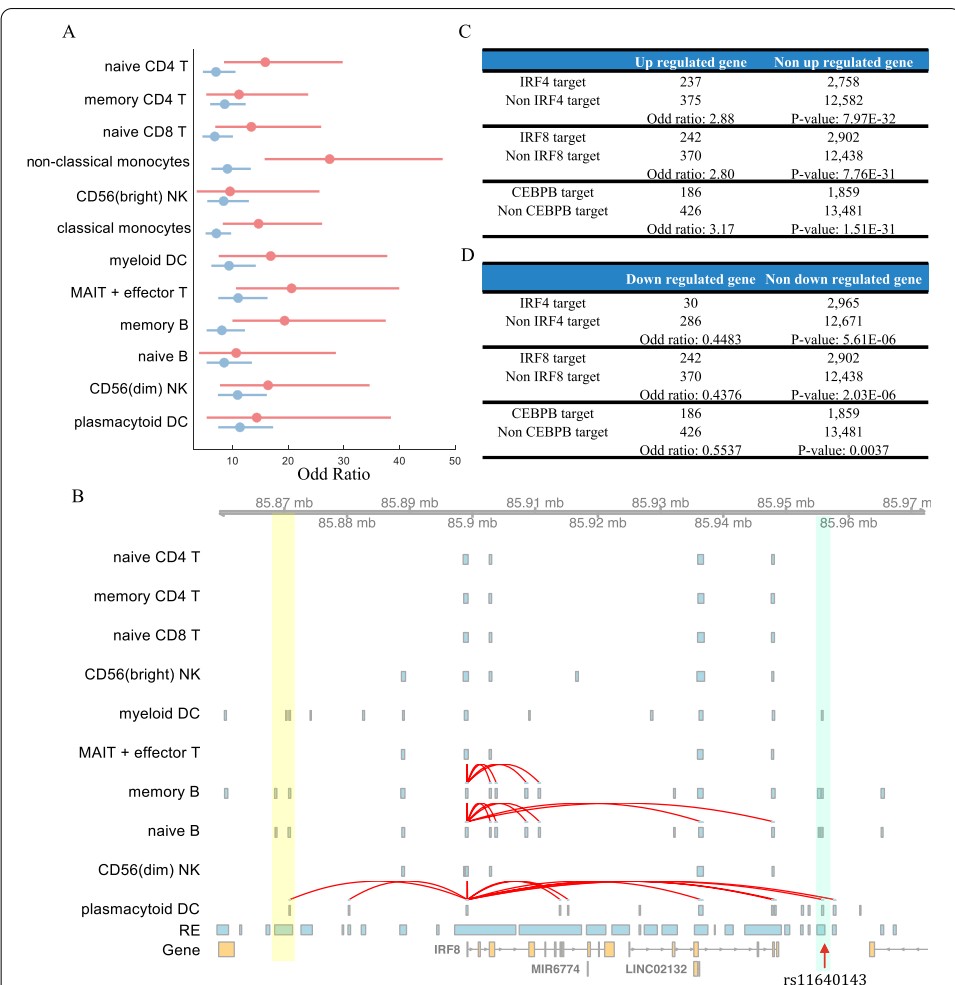

**Fig. 5** scREG interprets disease associated loci. **A** Comparison of the enrichment of fine-mapped GWAS variants of IBD disease in the scREG predicted regulatory elements (red) against that in all peaks (blue) called by MACS2 in each subpopulations. **B** Interactive visualization function of scREG package. scREG will plot genes, the raw peaks from all cells, peaks of each cluster, and the predicted peak-gene association in each cluster. Figure shows the track around the variant rs11640143 and gene IRF8. The highlighted blue bar represents the location of REs that contain the variant rs11640143, and the yellow bar represents one example of different cluster-level-peaks are merged into one in the raw peak calling before clustering. **C, D** Comparison of the three TFs (IRF4, IRF8, and CEBPB) target genes with the upregulated genes and downregulated genes in IBD patients. The *p*-value and odd ratio are calculated based on Fisher's exact test

Fig. 5B). The regulatory element that contains this variant is predicted to regulate IRF8 only in the scREG generated a *cis*-regulatory network of plasmacytoid DC cells. It also shows that the cluster level peaks are narrower than the raw peaks and multiple cluster-level peaks are merged into one raw peak (highlighted yellow in Fig. 5B). The interactive visualization function will help users understand the regulatory relations in a cell type-specific manner.

To further investigate the roles of these transcription factors in IBD, we modify our previous PECA2 method to infer their trans-regulatory target genes (see the "Inference of trans-regulatory targets" section) in each subpopulation. We download the differential expression genes list from IBD patients versus healthy controls study [44] for further

analysis. First, we find the TF IRF4 is upregulated in IBD patients compared to healthy controls (two samples two-tail *t*-test, *p*-value = 7.59E−31). Next, we compare differential expressed genes with the target genes of the three TFs (Fig. 5C, D). Target genes of IRF4 and IRF8 in plasmacytoid dendritic cells are 2.88 and 2.80 fold enriched (odds ratio) in the upregulated genes in IBD (Fisher's exact test, *p*-value 7.97E−32, and 7.76E−31). Target genes' of CEBPB in non-classical monocytes are 3.17-fold enriched in the IBD upregulated genes (Fisher's exact test, *p*-value 1.51E−31). Interestingly, downregulated genes in IBD are depleted in the IRF4, IRF8, and CEBPB's target genes (Odds ratios are 0.45, 0.44, and 0.55, *p*-values are 5.61E−06, 2.03E−06, and 0.0037). Overall, scREG is a useful tool for the interpretation of disease-associated loci.

## Discussion

Our method scREG uses a different strategy from Seurat V4 [18] to integrate the two types of data. In Seurat, the analysis consists of three main steps: (1) perform dimension reduction of each type of data separately, (2) calculate the distance matrix for each type of data based on the corresponding reduced dimension representation, and (3) merge the expression-based distance and the accessibility-based distance matrix into a single distance matrix via a weighted-nearest neighbor computation. In scREG, the dimension reduction is obtained through a joint optimization based on both types of data. One advantage of such analysis is that the integration of information from both types of data for dimension reduction allows us to make use of the multi-omics nature of the data to capture the real signal in an earlier stage of the analysis. The second advantage is that it provides a good input for those powerful existing methods [17, 45] to perform comprehensive analyses such as trajectory/pseudotime analysis. Last, we discuss the limitation of our analysis. To evaluate scREG method, we use the manually annotated cell labels. There is no guarantee that this label is 100% correct so that it may affect the results.

## Conclusions

In this paper, we proposed a computational method and developed an R package for the comprehensive analysis of single cell multiome gene expression and chromatin accessibility data. To analyze this type of bi-modality data, we propose a joint dimension reduction method. The reduced dimension representation is used for subpopulation identification and 2D embedding. Test results on four different datasets suggest that scREG is a useful and robust tool for cell population identification. The *cis*-regulatory potential defined in this paper provides direct information on subpopulation-specific regulatory networks. Validation of these networks was obtained by comparison with eQTL and 3D chromosome contact data. Finally, we applied our method to interpret the disease-associated loci of IBD and identified three key regulators.

## Methods

### Joint dimension reduction

Here, we introduce a dimension reduction method of single cell multiome data. The inputs are scATAC-seq data log2(1 + x) transformed count matrix $O$ ($p_1$ regions by $n$ cells matrix) and scRNA-seq data log2(1 + x) transformed count matrix $E$ ($p_2$ genes by $n$ cells matrix). The *cis*-regulatory information could be learned from the co-expression pattern

of RE accessibility and gene expression. This regulatory information is not included in either scRNA-seq or scATAC-seq, so it should be used for dimension reduction. First, we define a cell-level *cis*-regulatory potential $R_{ijc}$ for the *i*th RE and the *j*th TG in the *c*th cell as the sum of gene expression and RE accessibility weighted by a distance function $R_{ij} = (O_{ic} + E_{jc})e^{-d_{ij}/d_0}$, where $O_{ic}$ represents the accessibility of the *i*th RE in the *c*th cell, $E_{jc}$ represents the expression of *j*th TG in the *c*th cell, $d_{ij}$ represents the distance of *i*th RE to the *j*th TG, and the base parameter $d_0$ reflects the scale over which the weight decreases with distance (default value is 200 kb). To decrease the number of RE-TG pairs in the R matrix, we only consider RE-TG pairs with strong associations. Specifically, for a given RE-TG pair, we divide the cells into two groups based on the accessibility of the RE (zero v.s. non-zero) and perform a two-sample *t*-test. We select the top 10,000 RE-TG pairs based on the absolute value of the *t* statistics. After we obtain the R matrix, we do a term frequency-inverse document frequency transformation for the binarized O matrix [46]. We first calculate a log-scaled "term frequency" by dividing the accessibility of each cell by log(1 + total number of peaks accessible in the cell). We then multiplied the log-scaled "term frequency" by log scale "inverse document frequency", which is a peak-level index and calculated as log(1 + total number of cell/total number of cells in which the peak is accessible);

We do joint dimension reduction by an NMF-based optimization model. A dimension reduction of scATAC-seq can be obtained from a nonnegative matrix factorization $O = W_1 H$ as follows: the *i*th column of $W_1$ gives the *i*th base vector, while the *j*th column of $H$ gives the low dimensional representation of the *j*th cell. Similarly, the dimension reduction of the scRNA-seq and *cis*-regulatory potential can be obtained from the factorizations $E = W_2 H$ and $R = W_3 H$. These factorizations are obtained by solving the following optimization problem:

$$\min_{W_1, W_2, W_3, H} \frac{1}{2} \| O - W_1 H \|_F^2 + \frac{\lambda_1}{2} \| E - W_2 H \|_F^2 + \frac{\lambda_2}{2} \| R - W_3 H \|_F^2$$
$$\text{s.t.} \quad w_3^{ijk} = \left( w_1^{ik} + w_2^{jk} \right) e^{-d_{ij}/d_0} \tag{1}$$
$$W_1, W_2, W_3, H \geq 0;$$

There are three tuning parameters: $\lambda_1, \lambda_2$, and dimension of the H matrix K. We design an algorithm for this optimization problem.

$$w_1^{ik} \leftarrow w_1^{ik} \frac{\left( OH^T \right)^{ik}}{\left( W_1 HH^T \right)^{ik} + \varepsilon}$$

$$w_2^{jk} \leftarrow w_2^{jk} \frac{\left( EH^T \right)^{jk}}{\left( W_2 HH^T \right)^{jk} + \varepsilon}$$

$$w_3^{ijk} \leftarrow w_3^{ijk} \frac{\left( RH^T \right)^{ijk} + \left( w_1^{ik} + w_2^{jk} \right) e^{-d_{ij}/d_0}}{\left( W_3 HH^T + W_3 \right)^{ijk} + \varepsilon}$$

$$h_{kc} \leftarrow h_{kc} \frac{\left( W_1^T O + \lambda_1 W_2^T E + \lambda_2 W_3^T R \right)^{kc}}{\left( W_1^T W_1 H + \lambda_1 W_2^T W_2 H + \lambda_2 W_3^T W_3 H \right)^{kc} + \varepsilon}$$

The model is not too sensitive to the parameter K, so we can treat it as a fixed parameter, and the default value is 100. To choose the two tuning parameters $\lambda_1$ and $\lambda_2$, we introduce two interpretable parameters $\alpha$ and $\beta$.

$$\lambda_1 = \beta \frac{mean(W_{10}^T O)}{mean(W_{20}^T E)}$$

$$\lambda_2 = \alpha \frac{mean(W_{10}^T O)}{mean(W_{30}^T R)}$$

where $W_{10}$, $W_{20}$, $W_{30}$ are the solution of standard NMF after normalizing the square sum of H matrix to 1; Parameter $\alpha$ and $\beta$ are non-negative and reflect the ratio of regulatory potential and expression compared to chromatin accessibility; The default value of parameter $\alpha$ and $\beta$ is 1, which means we treat gene expression, chromatin accessibility, and the regulatory potential are equally important.

To test the effectiveness of the *cis*-regulatory potential term, we build a baseline model. We use the same strategy as described above to choose the tuning parameter $\lambda_1$. This model has the same objective function as intNMF [35]. Thus, we call this baseline method as intNMF. Please note that we use the same preprocessing and parameter selection strategy with scREG rather the strategies suggested in the intNMF paper.

$$\min_{W_1, W_2, H \geq 0} \frac{1}{2} \| O - W_1 H \|_F^2 + \frac{\lambda_1}{2} \| E - W_2 H \|_F^2$$

**Clustering of cells**

First, we calculate a cosine similarity between each pair of cells based on the reduced dimension matrix H. It is hard to detect the rare populations based on this similarity matrix because (1) rare populations have a small proportion in the total cost and (2) the noise would spuriously link unrelated parts of the graph. To increase the weight of the rare population and to denoise the similarity matrix, we refine the similarity of cells based on their shared nearest neighbors. Specifically, we extract the nearest C neighbors for each cell and calculate the Jaccard index for each pair of cells based on their nearest C neighbors. The Jaccard index uses the local density at each cell to remove spurious edges and strengthen well-supported pairs [47]. Here, the default value of C is the square root of the number of cells. We cluster the cells based on the Louvain algorithm [37]. To test the usefulness of the cis-regulatory term, we use the same clustering method as scREG on the H matrix from intNMF method to cluster the cell.

**Construction of the cis-regulatory networks**

The $W_3$ matrix reflects the regulatory activity of RE-TG pairs on each of the K dimensions. Note that the number of dimension K is different from the number of clusters. To extract the *cis*-regulatory score (CRS) of each cluster, we multiply $W_3$ by cluster means of the H matrix.

$$CRS = W_3 h$$

where *h* represents the cluster mean profile of the H matrix. The $CRS_{ijk}$ represent the *cis*-regulatory score of *i*th TF and *j*th TG on *k*th cluster. Since the CRS is computed on predefined RE-TG pairs which have correlation across cells, a higher CRS in a subpopulation indicates a potential cis-regulation. For each cluster, we output the top 10,000 RE-TG pairs as the *cis*-regulatory network. The REs used in this network are peaks called from all merged cells. To increase the sensitivity, we merge the cells from the same cluster and call peaks by MACS 2[48]. We overlap the MACS2 peaks with the RE-TG pairs from scREG to obtain more accurate *cis*-regulatory networks.

### Joint embedding of scRNA-seq and scATAC-seq

Our method does linear transformations for scATAC-seq and scRNA-seq to get the H matrix, which is in a common K dimension space. Based on the normalized (column square sum to one) H matrix, we use cosine distance and reduce them from K dimensions to two dimensions by tSNE and UMAP.

### Model parameters

We have several parameters for scREG package, and all of them have default values. For all analysis in this manuscript, we use the default parameters. The number of RE-TG pairs used for the construction of the R matrix has a default value of 10000. The dimension of the H matrix has default value of 100. The hyper parameters $\alpha$ and $\beta$ have default value of 1. Parameter C, the number of nearest neighbors used in the clustering analysis, has default value of square root of number of cells. For scAI method, we use following parameters "K = 100, nrun = 1, do.fast = T." Here, we use 100 dimensions to make it consistent with the other methods. For MOFA+ analysis, we followed the tutorial of MOFA website https://raw.githack.com/bioFAM/MOFA2_tutorials/master/R_tutorials/10x_scRNA_scATAC.html. Only difference is that we set $K = 100$ to make it consistent with other methods. For Seurat, we follow the tutorial of Seurat website https://satijalab.org/seurat/articles/weighted_nearest_neighbor_analysis.html.

### Evaluation metrics

To evaluate the dimension reduction, we use silhouette index [49] for 14-dimension matrix for PBMC data. For PCA and LSA, we directly reduce the dimension to 14. For scAI, MOFA, and scREG, we reduce dimension to 100 and further reduce it to 14 by PCA. For clustering evaluation, we use NMI [50] and ARI [51] to compare the clustering label and ground truth. For internal clustering evaluation, we use Callnski-Harabasz index [52], Davies-Bouldln index [53], Silhouette index, and Modularity Q [54]. The inputs are clustering label and cell-cell distance matrix (calculated on the 2D embedding or top 20 PCs) on gene expression or chromatin accessibility.

### PBMC 10 K data

We download the PBMC 10 K data from the 10X genomics website https://support.10xgenomics.com/single-cell-multiome-atac-gex/datasets. Note that it contains 11,909 cells, and

the granulocytes were removed by cell sorting of this dataset. We use the filtered cells by features matrix from the output of 10X genomics software Cell Ranger ARC as input and perform the downstream analysis. First, we perform Seurat 4.0 weighted nearest neighbor (WNN) analysis [18], and it removes 1497 cells. We also remove the cells that don't have surrogate ground truth [34], and it results in 9543 cells.

### Enrichment of GWAS variants

To perform the enrichment analysis, we divide variants into four groups ($2\times2$) based on two categories: significant or not significant, and located in regulatory element or not. Total variants used here consist of significant variants and background variants. The 10 million background variants are downloaded from the stratified linkage disequilibrium score regression (S-LDSC) website [43], which is defined as variants in the 1000 Genomes Project with minor allele count $>5$ in 379 European samples. We calculate the odds ratios of enrichment based on the $2\times2$ table.

$$Odd\ Ratios = \frac{\#of\ significant\ variants\ in\ RE / \#of\ significant\ variants}{\#of\ background\ variants\ in\ RE / \#of\ background\ variants}$$

### Inference of trans-regulatory targets

Our previous method PECA takes paired expression and chromatin accessibility data across diverse cellular contexts as input, models how trans- and *cis*-regulatory elements work together to affect gene expression in a context-specific manner, and output the transcriptional regulatory network with TF-RE-TG as the building block [55]. PECA2 aims to infer the regulatory network in a new cellular context different from those used in training the model by selecting active REs, specifically expressed TFs and expressed TGs in this context [36]. The trans-regulation score (TRS) between given $i$th TF and $j$th TG is defined as follows.

$$TRS_{ij} = \left(\sum_k B_{ik}\widetilde{RE}_k I_{kj}\right) \times 2^{|R_{ij}|} \times \sqrt{\widetilde{TF_i}\widetilde{TG_j}}$$

where $B_{ik}$ is motif binding strength of $i$th TF on $k$th RE, which is defined as the sum of binding strength (motif position weight matrix-based log-odds probabilities, see HOMER software for detail) of all of the binding sites on this RE; $\widetilde{RE}_k$ represent the normalized accessibility ($RE_k \times \frac{RE_k}{median(RE_k)}$) of $k$th RE. The first term $RE_k$ represents the actual accessibility of the RE, and the second term represents relative accessibility compared to the median accessibility level of this RE on external data. If one RE is accessible in the given cellular context, and the accessibility is also much higher than the accessibility level on other contexts, then this RE is specifically accessible in given cellular context; $I_{kj}$ represents the interaction strength between $k$th RE and $j$th TG, which is the CRS score between $k$th RE and $j$th TG in this paper. $\widetilde{TG_j}$ and $\widetilde{TF_i}$ represents the normalized expression level of $j$th TG ($TG_j \times \frac{TG_j}{median(TG_j)}$) and $i$th TF respectively. $R_{ij}$ is the expression correlation of $i$th TF and $j$th TG across diverse cellular contexts. Higher regulation score $TRS_{ij}$ implies $j$th TG is more likely to be regulated by $i$th TF.

## Supplementary Information

---

**Additional file 1: Figure S1**. Comparison of SI resulting from different methods. **Figure S2**. UMAP visualization of scAI and MOFA+ clustering. **Figure S3**. Performance of different Methods under different resolutions. **Figure S4**. UMAP visualization of marker genes' expression on mouse E18 brain data. **Figure S5**. UMAP visualization of marker genes' expression on human cerebellum data. **Figure S6**. UMAP visualization of marker genes' expression on lymph node from B cell lymphoma data. **Figure S7**. Evaluation of Seurat V4 clustering and scREG clustering on Cell Ranger scRNA-seq embedding and Cell Ranger scATAC-seq embedding on mouse E18 brain. **Figure S8**. Evaluation of Seurat V4 clustering and scREG clustering on Cell Ranger scRNA-seq embedding and Cell Ranger scATAC-seq embedding on human cerebellum. **Figure S9**. Evaluation of Seurat V4 clustering and scREG clustering on Cell Ranger scRNA-seq embedding and Cell Ranger scATAC-seq embedding on lymph node from B cell lymphoma. **Figure S10**. UMAP visualization of BMMC cell using Seurat and scREG clustering. **Figure S11**. Validate the RE-TG prediction by HiC data. **Figure S12**. AUROC and AUPR predicting HiC data. **Figure S13**. Clusters specificity of CRS.

**Additional file 2.** Supplementary note for evaluation of embedding when the ground truth is not available.

**Additional file 3.** Review history.

---

### Peer review information

Barbara Cheifet and Kevin Pang were the primary editors of this article and managed its editorial process and peer review in collaboration with the rest of the editorial team.

### Review history

The review history is available as Additional file 3.

### Authors' contributions

Z.D and W.H.W. conceived the project. Z.D. designed the analytical approach and performed the data analysis with the help of F.N., J.X., and Q. L. Z.D. and F.C. wrote the software. Z.D., W.H.W., and F.N. wrote, revised, and contributed to the final manuscript. The authors read and approved the final manuscript.

### Funding

This work was partially supported by NIH grants P20 GM139769, R01 HG010359, and P50 HG007735.

### Availability of data and materials

The R package of scREG is available on Github at https://github.com/Durenlab/RegNMF [56] and zendo repository under the GPLv3 license [57]. The multiome data is from the 10X genomics website at https://support.10xgenomics.com/single-cell-multiome-atac-gex/datasets. Bone marrow mononuclear cell data are from the NeurIPS competition training data https://openproblems.bio/neurips_docs/data/. The fine mapping 376 variants of IBD are downloaded from GitHub https://github.com/EngreitzLab/ABC-GWAS-Paper/blob/main/ABC-Max/out/ABC/IBD/IBD.sig.varList.tsv. The background variants are downloaded from the LDSCORE website model v1.1 https://alkesgroup.broadinstitute.org/LDSCORE/baselineLD_v1.1_hg38_annots/.

## Declarations

### Ethics approval and consent to participate
Not applicable.

### Consent for publication
Not applicable.

### Competing interests
The authors declare no competing interests.

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

## 
