## [**Additional file 3.** Review history. · Genome Biology]

Review History

First round of review

Reviewer 1

Were you able to assess all statistics in the manuscript, including the appropriateness of statistical tests used? Yes, and I have assessed the statistics in my report.

Were you able to directly test the methods? No.

Comments to author:

The authors presented scREG, a multi-task tool that takes jointly profiled single cell gene expression and chromatin accessibility data as input, and performs dimension reduction, clustering, inference of cis-regulatory relationship between genomic regions and genes, and inference of gene regulatory networks. The model specifically considers cis-regulatory relationship during data integration, which is an advantageous feature compared to other tools that perform similar tasks using the same types of input data. Overall this is a novel and interesting approach aiming to tackle a significant and timely problem in single cell multi-omic data integration and analysis. My concerns with respect to the current form of the manuscript are as follows:

Major concerns:

1. The authors compared multiple dimension reduction methods to show the advantage of scREG in terms of identifying cell populations. A majority of the baseline methods are very basic and naive methods, such as, NMF ATAC, PCA ATAC, NMF RNA and PCA RNA. More sophisticated methods have been developed in the past to integrate matched (jointly profiled) scRNA-seq and scATAC-seq data, see [Argelaguet, R., Cuomo, A. S. E., Stegle, O. & Marioni, J. C. Computational principles and challenges in single-cell data integration. Nat. Biotechnol. 1-14 (2021)]. Among these methods, I think the authors should compare scREG with scAI [Jin, S., Zhang, L. & Nie, Q. scAI: an unsupervised approach for the integrative analysis of parallel single-cell transcriptomic and epigenomic profiles. Genome Biol. 21, 25 (2020)] in terms of dimension reduction and subpopulation detection, which is also based on a matrix factorization framework, but without using the cis regulatory potential term in scREG.
2. The cis-regulatory scores predicted for each peak-gene pair in each cluster seems to be better interpreted as an indicator of how much that peak-gene pair contributes to that cluster, as opposed to how strong the regulatory relationship is in that peak-gene pair in the cluster. If the authors meant cis-regulatory scores to indicate the strength of the peak-gene regulatory with the cis-regulatory scores, please explain theoretically why.
3. Related to point 2, can the authors please explain why the equation of calculating the cis-regulatory potential $R_{(ij)c}$ is designed to the specific form. For example, why O_{ic} is summed up with E_{jc} instead of multiplying the two terms. Also, no normalization on O or E is mentioned in Methods for this calculation: since O has binary values, what if the E values are

large and dominate the O values?

Minor concerns:

1. Please specify whether all peak-gene pairs are used when constructing matrix R. If they are selected, what are the criteria for selection?
2. Comparing the cis-regulatory scores predicted with scREG with the eQTL or GWAS data involves determining whether a region in the ATAC-seq data is the same region as that in eQTL or GWAS data. How was that determined?
3. It is not clear what the last sentence in Discussion (lines 29-30) means: can the authors give some examples of "those powerful existing methods" and what kind of "comprehensive analyses"?
4. The CRS scores are cluster specific which indicates that different clusters have different top peak-gene pairs. Considering that most of the existing methods which infer peak-gene relationship do not consider difference between cell types, it would be interesting to see quantitative results on how much difference is in the top peak-gene pairs in different clusters.
5. Please include details on how the enrichment results in Fig. 5A are obtained.

Reviewer 2

Were you able to assess all statistics in the manuscript, including the appropriateness of statistical tests used? Yes, and I have assessed the statistics in my report.

Were you able to directly test the methods? No.

Comments to author:

The authors propose a dimension reduction methodology, based on the concept of cis-regulatory potential, for single cell multiome gene expression and chromatin accessibility data.

Major points:

1. Integrative dimensionality reduction methods based on NMF is not a novelty of this work. It already exist and has been applied in genomics (for example intNMF: <https://journals.plos.org/plosone/article?id=10.1371/journal.pone.0176278>). The only novelty of the dimensionality reduction performed in scREG is the fact of adding in the loss a term related to cis-regulatory information. This point should be clarified and existing works performing the same integrative NMF should be cited.

* In addition, the authors state that "Preserving the cis-regulatory information should be an important requirement of the dimension reduction step". This is a really interesting aspect, that however the authors do not test in their work. It would thus be fundamental to test the effect of this additional term on the dimensionality reduction by comparing intNMF, or equivalent

approaches, with respect to scREG.

2. There is a general issue in the dimensionality reduction testing (Figure 2 and Figure 3), scREG designed for multi-omics dimensionality reduction is always tested by the authors with respect to methods designed for dimensionality reduction in single omics (with the only exception of Seurat 4). This is quite strange, as other methods for multi-omics dimensionality reduction exist and comparing with them would be more pertinent. I see the results in Figure 2 as a way to compare the added value of multi-omics over single omics, but this does not imply that scREG is better than existing dimensionality reduction techniques.

3. On the other side, the authors never propose a proper benchmarking of scREG over existing multi-omics dimensionality reduction approaches. In Figure 3 still methods designed for single omics are considered, with the only exception of Seurat4. Methods designed for multi-omics should be the focus of Figure4, including for example MOFA+, scMM, scMVAE, plus Seurat4 (already included in the comparison).

* In addition, performances in one dataset are not sufficiently exhaustive, but multiple datasets should be taken into account for a proper evaluation.

4. Still in line with the previous point, in PBMCs how is the ground truth defined? The authors state "we used the cell-type labels that were annotated by the 10X Genomics R&D team as surrogates for ground truths[35]". How the 10X Genomics R&D team annotated the labels? Based on FACS sorting? Based on scRNA-seq markers? Or other? This is a crucial point as, if the markers are based on scRNAseq markers than a method driven by scRNAseq will perform better, just because of a bias in the evaluation.

* What the authors mean with "as surrogates for ground truths[35]" ? this should be better explained

* Finally, combining NMI and ARI with a label-free evaluation is here fundamental as no universally correct label exist in single-cell data. Some cells classified as B cells might instead belong to a rare population that ended up to be under detected, for example.

5. Regarding the cis-regulatory networks then, I find to be a missing element the fact of comparing the networks that can be inferred by scREG with the existing network inference approaches such as SCENIC. Benchmarking platforms also exist and could help such comparison <https://github.com/Murali-group/Beeline>

6. Finally, regarding the scREG tool, to assure reproducibility in the long timing and a proper maintenance a submission of the R package to CRAN or Bioconductor is fundamental.

Reviewer #1:

The authors presented scREG, a multi-task tool that takes jointly profiled single cell gene expression and chromatin accessibility data as input, and performs dimension reduction, clustering, inference of cis-regulatory relationship between genomic regions and genes, and inference of gene regulatory networks. The model specifically considers cis-regulatory relationship during data integration, which is an advantageous feature compared to other tools that perform similar tasks using the same types of input data. Overall this is a novel and interesting approach aiming to tackle a significant and timely problem in single cell multi-omic data integration and analysis. My concerns with respect to the current form of the manuscript are as follows:

Response: *Thank you for your time to review and your positive comments.*

Major concerns

Comment 1: The authors compared multiple dimension reduction methods to show the advantage of scREG in terms of identifying cell populations. A majority of the baseline methods are very basic and naive methods, such as, NMF ATAC, PCA ATAC, NMF RNA and PCA RNA. More sophisticated methods have been developed in the past to integrate matched (jointly profiled) scRNA-seq and scATAC-seq data, see [Argelaguet, R., Cuomo, A. S. E., Stegle, O. & Marioni, J. C. Computational principles and challenges in single-cell data integration. *Nat. Biotechnol.* 1-14 (2021)]. Among these methods, I think the authors should compare scREG with scAI [Jin, S., Zhang, L. & Nie, Q. scAI: an unsupervised approach for the integrative analysis of parallel single-cell transcriptomic and epigenomic profiles. *Genome Biol.* 21, 25 (2020)] in terms of dimension reduction and subpopulation detection, which is also based on a matrix factorization framework, but without using the cis regulatory potential term in scREG.

Response: *Thank you for your instructive comments. We agree reviewer that scAI is a great method to compare. In this resubmission, we added the comparison of our method with scAI and MOFA+. The results in Fig. 2B shows that our method outperforms both scAI and MOFA+. These results confirmed that preserving the cis-regulatory potential in dimension reduction is useful.*

Comment 2: The cis-regulatory scores predicted for each peak-gene pair in each cluster seems to be better interpreted as an indicator of how much that peak-gene pair contributes to that cluster, as opposed to how strong the regulatory relationship is in that peak-gene pair in the cluster. If the authors meant cis-regulatory scores to indicate the strength of the peak-gene regulatory with the cis-regulatory scores, please explain theoretically why.

Response: *Thank you for your comments. The cis-regulatory scores are defined on predefined RE-TG pairs which have a significant correlation across cells. Thus, those predefined RE-TG may reflect cis-regulation in some cell populations. To find in which cell population do the RE-TG has regulation, we define the cis-regulatory scores. A higher cis-regulatory score in one cell population means RE and gene are expressed in this population. So we can interpret a RE-TG pair with a high cis-regulatory score as potential cis-regulation in the given cell population. We added the following sentence to the Method section*

“Since the CRS is computed on predefined RE-TG pairs which have correlation across cells, a higher CRS in a subpopulation indicate a potential cis-regulation.”

Comment 3: Related to point 2, can the authors please explain why the equation of calculating the cis-regulatory potential $R_{(ij)c}$ is designed to the specific form. For example, why O_{ic} is summed up with E_{jc} instead of multiplying the two terms. Also, no normalization on O or E is mentioned in Methods for this calculation: since O has binary values, what if the E values are large and dominate the O values?

Response: Thank you for your comments. As described in the Method section, we do $\log_2(1+x)$ transformation for gene expression data and do a frequency-inverse document frequency transformation (TF-IDF) for chromatin accessibility data. After those transformations, the gene expression and chromatin accessibility data have a similar scale. We have tried to use multiplication instead of the sum, the results become worse. We think the reason is that some values in the R matrix become very large. If we do not do the TF-IDF transformation, constructing the R matrix by multiplying the two terms works well. But we decided to use the sum instead of multiplication considering the fact that the TF-IDF transformation increases the clustering accuracy significantly, which is consistent with previous literature (<https://divingintogeneticsandgenomics.rbind.io/post/clustering-scatacseq-data-the-tf-idf-way/>).

Minor concerns

Comment 4: Please specify whether all peak-gene pairs are used when constructing matrix R. If they are selected, what are the criteria for selection?

Response: Thank you for your comments. Yes, we selected some RE-TG pairs based on their co-expression pattern across cells. We add following description in the Method section.

"To decrease the number of RE-TG pairs in the R matrix, we only consider RE-TG pairs with strong association. Specifically, for a given RE-TG pair, we divide the cells into two groups based on the accessibility of the RE (zero v.s. non-zero) and perform two-sample t-test. We select top 10,000 RE-TG pairs based on the absolute value of the t statistics."

Comment 5: Comparing the cis-regulatory scores predicted with scREG with the eQTL or GWAS data involves determining whether a region in the ATAC-seq data is the same region as that in eQTL or GWAS data. How was that determined?

Response: Thank you for your comments. For eQTL validation, we use RE-TG pairs from the R matrix as the basis. If the RE contains any SNPs that have a significant association with the TG from eQTL, then we take such RE-TG pair as a positive sample otherwise negative sample. We do not consider the SNPs located outside the REs.

Comment 6: It is not clear what the last sentence in Discussion (lines 29-30) means: can the authors give some examples of "those powerful existing methods" and what kind of "comprehensive analyses"?

Response: Thank you for your comments. Many methods take the reduced dimension matrix as input, for example, trajectory/ pseudotime analysis by Slingshot or Monocle. We have added an explanation in this version.

Comment 7: The CRS scores are cluster specific which indicates that different clusters have different top peak-gene pairs. Considering that most of the existing methods which infer peak-

gene relationship do not consider difference between cell types, it would be interesting to see quantitative results on how much difference is in the top peak-gene pairs in different clusters.

Response: Thank you for your comments. Considering your suggestions, we calculate Jaccard similarity between clusters based on top 10,000 peak-gene pairs with highest CRS score in each cluster. Supplementary Figure S12 shows that the average Jaccard similarity between clusters is 0.47. The difference is dependent on the similarity between cell types. In naïve B cell, about 11.45% of the peak-gene pairs are different from that in the memory B cells. However, about 51.78% of peak-gene pairs in naïve B cells are different from that in the classical monocytes. The following description is added to the end of "scREG constructs subpopulation specific gene regulatory networks" section.

"The cis-regulatory networks are highly cell-type-specific. Supplementary Figure S11 shows the Jaccard similarity of clusters in terms of cis-regulation. The average Jaccard similarity between clusters is 0.4760. Hierarchical clustering analysis shows similar cell types have a similar cis-regulatory network. For example, the average similarity between four T cell clusters is 0.7783, and group to one cluster; the similarity of two B cell subpopulations is 0.79; the similarity of two NK cell subpopulations is 0.79; similarity of two monocytes subpopulation is 0.64."

Comment 8: Please include details on how the enrichment results in Fig. 5A are obtained.

Response: Thank you for your comments. In light of your suggestion, we add a section in the Methods to describe the details of the enrichment analysis.

"To perform the enrichment analysis, we divide variants into four groups (2×2) based on two categories: significant or not significant, and located in regulatory element or not. Total variants used here consist of significant variants and background variants. The 10 million background variants are downloaded from the stratified linkage disequilibrium score regression (S-LDSC) website [41], which is defined as variants in the 1000 Genomes Project with minor allele count >5 in 379 European samples. We calculate the odds ratios of enrichment based on the 2×2 table.

$$\text{Odd Ratios} = \frac{\# \text{ of significant variants in RE} / \# \text{ of significant variants}}{\# \text{ of background variants in RE} / \# \text{ of background variants}}$$

"

Reviewer #2

The authors propose a dimension reduction methodology, based on the concept of cis-regulatory potential, for single cell multiome gene expression and chromatin accessibility data.

Response: Thank you for your time to review.

Major points:

Comment 1: Integrative dimensionality reduction methods based on NMF is not a novelty of this work. It already exist and has been applied in genomics (for example intNMF: <https://journals.plos.org/plosone/article?id=10.1371/journal.pone.0176278>). The only

novelty of the dimensionality reduction performed in scREG is the fact of adding in the loss a term related to cis-regulatory information. This point should be clarified and existing works performing the same integrative NMF should be cited.

* In addition, the authors state that "Preserving the cis-regulatory information should be an important requirement of the dimension reduction step". This is a really interesting aspect, that however the authors do not test in their work. It would thus be fundamental to test the effect of this additional term on the dimensionality reduction by comparing intNMF, or equivalent approaches, with respect to scREG.

Response: Thank you for your comments. We have added a description of NMF based methods to clarify the contribution of other research and novelty of this research, and we also have cited intNMF in this description.

"Based on the CRP concept, we designed a non-negative matrix factorization (NMF) based optimization model to project the cells into a lower dimension space. Several NMF based methods [3, 4, 36, 37] have been developed for dimension reduction and clustering of single cell genomics data and have shown great advantages in data integration. In addition to the gene expression matrix and chromatin accessibility matrix, we also include the cis-regulatory potential matrix as one input of the optimization model to use a lower dimension to represent a cell."

To demonstrate the usefulness of the additional cis-regulatory potential term, we have built a baseline model in which the cis-regulatory potential term was removed, and we compared our method with this baseline to show the usefulness of the cis-regulatory potential. In addition to the scREG_baseline method, we also added two NMF based methods, scAI and MOFA+, for comparison in this resubmission. The results in Fig. 2B shows that our method outperforms both scAI and MOFA+. These results illustrate that preserving the cis-regulatory potential in dimension reduction is useful.

Comment 2: There is a general issue in the dimensionality reduction testing (Figure 2 and Figure 3), scREG designed for multi-omics dimensionality reduction is always tested by the authors with respect to methods designed for dimensionality reduction in single omics (with the only exception of Seurat 4). This is quite strange, as other methods for multi-omics dimensionality reduction exist and comparing with them would be more pertinent. I see the results in Figure 2 as a way to compare the added value of multi-omics over single omics, but this does not imply that scREG is better than existing dimensionality reduction techniques.

Response: Thank you for your comments. In this resubmission, we added comparisons of our method with the other 3 multi-omics methods. Please see Fig. 2B and the response to comment 1 for detail. These results show that scREG is better than existing dimension reduction methods as it is preserving the cis-regulatory potential information in the dimension reduction.

Comment 3: On the other side, the authors never propose a proper benchmarking of scREG over existing multi-omics dimensionality reduction approaches. In Figure 3 still methods designed for single omics are considered, with the only exception of Seurat4. Methods designed for multi-omics should be the focus of Figure4, including for example MOFA+, scMM, scMVAE, plus Seurat4 (already included in the comparison).

* In addition, performances in one dataset are not sufficiently exhaustive, but multiple datasets should be taken into account for a proper evaluation.

Response: Thank you for your comments. In light of your suggestion, we compared our method with scAI and MOFA+ in terms of clustering. The results are presented in Fig. 3 and supplementary Figure S2. In the resubmission, our method has been compared with three multi-omics methods: scAI, MOFA+, and Seurat4, and our method outperforms all three methods in NMI and ARI based evaluation.

Comment 4: Still in line with the previous point, in PBMCs how is the ground truth defined? The authors state "we used the cell-type labels that were annotated by the 10X Genomics R&D team as surrogates for ground truths[35]". How the 10X Genomics R&D team annotated the labels? Based on FACS sorting? Based on scRNA-seq markers? Or other? This is a crucial point as, if the markers are based on scRNAseq markers than a method driven by scRNAseq will perform better, just because of a bias in the evaluation.

* What the authors mean with "as surrogates for ground truths[35]" ? this should be better explained

* Finally, combing NMI and ARI with a label-free evaluation is here fundamental as no universally correct label exist in single-cell data. Some cells classified as B cells might instead belong to a rare population that ended up to be under detected, for example.

Response: Thank you for your comments. We obtain the cell label from the tutorial of MOFA+ in the following website:

https://raw.githack.com/bioFAM/MOFA2_tutorials/master/R_tutorials/10x_scRNA_scATAC.html

The PBMC data we use here is released on the 10X Genomics website and these cell type annotations are also done by the 10X Genomics R&D team. As far as we know, this data is the only sc-multiome data with known labels. We agree reviewer that the cell annotation may contain some wrong annotations, especially for rare populations. To deal with this, in addition to NMI and ARI based evaluation, we also performed a label-free evaluation. In Fig. 3F, we compared our method with Seurat using 4 evaluation metrics: Calinski-Harabasz index, Davies-Bouldin index, Silhouette index, and Modularity Q. Both Label-based evaluation and label-free evaluation show that our method provides good clustering results. The main contribution of this paper is that the dimension reduction method we developed. Taking our reduced dimension matrix as input, one could user any clustering method to identify cell populations.

Comment 5: Regarding the cis-regulatory networks then, I find to be a missing element the fact of comparing the networks that can be inferred by scREG with the existing network inference approaches such as SCENIC. Benchmarking platforms also exist and could help such comparison <https://github.com/Murali-group/Beeline>

Response: Thank you for your comments. Here we focus on the construction of cis-regulatory networks, which means we are trying to connect cis-regulatory elements to target genes. While the SCENIC and the benchmarking paper Beeline are talking about the trans-regulatory network which connects transcription factors to target gene. Inference of cis-regulation requires integration of chromatin accessibility data with gene expression while the trans-regulation could be done just based on the gene expression data. Pearson correlation coefficient (PCC) is the method people use to infer cis-regulation from multiome data, for example, the chromatin accessibility package Signac (the same developer with Seurat) uses

PCC for cis-regulatory network construction. Thus, here we just compare our method to PCC based method.

Comment 6: Finally, regarding the scREG tool, to assure reproducibility in the long timing and a proper maintenance a submission of the R package to CRAN or Bioconductor is fundamental.

Response: *Thank you for your comments. We agree that submitting the R package to CRAN or Bioconductor is helpful for reproducibility and maintenance. We have a plan to submit the scREG package to Bioconductor, however, it requires a long time to prepare, and Bioconductor only has two release dates per year. So we plan to submit it to Bioconductor in the future. To increase the reproducibility, we have deposited the source code to a DOI-assigning repository named Zenodo and cited source code in the manuscript.*

Second round of review

Reviewer 1

I'd like to thank the authors for their effort in improving the manuscript based on reviewers' comments. My concerns were mostly addressed. I have one additional requests: could the authors please specify details in obtaining the results of scREG, scAI and MOFA+ in Fig2, including preprocessing of data, and choice of hyper-parameters (eg. the weights for different loss terms)?

Reviewer 2

I thank the authors for all the work done to address my comments. However, still some points should be addressed in the work.

Major points:

1. In the existing papers proposing methods for multi-omics single-cell data integration (see: <https://www.embopress.org/doi/full/10.15252/msb.20178124> and https://openproblems.bio/neurips_docs/about_tasks/task3_joint_embedding/) authors normally report the average silhouette index (SI) computed across all cells. The authors report the information regarding this in Supp Figure S1, but the average SI should be mentioned in the Results and discussed, to make the comparison comparable with existing papers.
2. The label-free evaluation performed in the paper makes no sense according to me. The authors are evaluating the quality of their clustering using the UMAP coordinates, which is a major error. Having cells close in the UMAP does not speak about the quality of the clustering performed on the 100-dimensional embedding. Another evaluation should be used.
3. The authors mention that they tested performances over four datasets. However, three out of four datasets they have no labels. Other datasets with labels should be used for the SI comparison.
4. The authors report performances of scREG_bl the version of scREG without cis-regulatory term. To my understanding, scREG_bl already exist as a method and it is called intNMF (<https://journals.plos.org/plosone/article?id=10.1371/journal.pone.0176278>). The authors should cite this method and call it with its name in their comparison.
5. The authors should provide in the methods all the details concerning the procedure followed in the paper. Namely, how they evaluated the quality of the clusters, with which scores and applied to what etc.
6. The authors talk in the paper about gene regulatory networks, but what they are really computing are cis-regulatory networks. The difference in between the two should be clarified and gene regulatory networks should not be used in the paper.

Minor points:

1. In response to my comment 1, the authors affirm that MOFA+ is a NMF-based method. This is not true. MOFA is based on Bayesian Factor Analysis.
2. Relative to my comment 4, the authors should discuss the limitations of their choice. Namely, using labels based on scRNA-seq data will associate better performances to methods that are more driven by scRNAseq in the integration. The labels provided by 10X are thus not perfect

labels for the evaluation of integrative methods, even if I agree that they are the only solution currently available.

Reviewer #1:

Comment: I'd like to thank the authors for their effort in improving the manuscript based on reviewers' comments. My concerns were mostly addressed. I have one additional requests: could the authors please specify details in obtaining the results of scREG, scAI and MOFA+ in Fig2, including preprocessing of data, and choice of hyper-parameters (eg. the weights for different loss terms)?

Response: Thank you for your time to review our manuscript again. In light of your suggestion, we have added this description in the Method section.

“Model parameters

We have several parameters for scREG package, and all of them have default values. For all analysis in this manuscript, we use the default parameters. The number of RE-TG pairs used for the construction of the R matrix has a default value of 10,000. The dimension of the H matrix has default value of 100. The hyper parameters α and β have default value of 1. Parameter C, the number of nearest neighbors used in the clustering analysis, has default value of square root of number of cells. For scAI method, we use following parameters “K = 100, nrun = 1, do.fast = T”. Here we use 100 dimensions to make it consistent with the other methods. For MOFA+ analysis, we followed the tutorial of MOFA website https://raw.githack.com/bioFAM/MOFA2_tutorials/master/R_tutorials/10x_scRNA_scATAC.html. Only difference is that we set K=100 to make it consistent with other methods. For Seurat we follow the tutorial of Seurat website https://satijalab.org/seurat/articles/weighted_nearest_neighbor_analysis.html. “

Reviewer #2:

Reviewer #2: I thank the authors for all the work done to address my comments. However, still some points should be addressed in the work.

Major points:

Comment 1. In the existing papers proposing methods for multi-omics single-cell data integration (see: <https://www.embopress.org/doi/full/10.15252/msb.20178124> and https://openproblems.bio/neurips_docs/about_tasks/task3_joint_embedding/) authors normally report the average silhouette index (SI) computed across all cells. The authors report the information regarding this in Supp Figure S1, but the average SI should be mentioned in the Results and discussed, to make the comparison comparable with existing papers.

Response: Thanks for your comments. A description of the average silhouette index on each cell type and a discussion has been added to results session and showed in the Supplementary Figure S1.

“Supplementary Figure S1 shows the comparison of the average SI on all the cell types by different dimension reduction methods. Except for effector CD8 cell, on which all methods generally have poor performance. Our method scREG performs better on most of the cell types, SI range from 0.24 to 0.98, and achieve the highest average SI across cell types (0.56)”

Comment 2. The label-free evaluation performed in the paper makes no sense according to me. The authors are evaluating the quality of their clustering using the UMAP coordinates, which is a major error. Having cells close in the UMAP does not speak about the quality of the clustering performed on the 100-dimensional embedding. Another evaluation should be used.

Response: *If cells with same label are close to each other and far from cells with different label for both gene expression and chromatin accessibility space, then we think this is a good clustering method. If the clustering is neither supported by gene expression nor supported by chromatin accessibility, then this is a bad clustering. Most internal clustering evaluation methods take as input a distance matrix among cells and a clustering label. So, the key problem becomes how to calculate the distance matrix.*

Since the original data is too sparse and affected by drop-out, the distance calculation on the raw data cannot obtain the relationship among cells. If we calculate this distance matrix based on the reduced dimension matrix from scREG, the result reflects the effectiveness of the Louvain clustering method rather than the scREG method. We agree that the distance calculation on Umap may not reflect the real distance. In this revised version, we also calculate distance based on top 20 principal components. The results are shown in Supplementary Figure S9 and which is mostly consistent with the previous Umap-based result.

Comment 3. The authors mention that they tested performances over four datasets. However, three out of four datasets they have no labels. Other datasets with labels should be used for the SI comparison.

Response: *Thanks for your comment. To address this problem, we have downloaded the training data set from NeurIPS "Open Problem in Single Cell Analysis" competition: The 10X Genomics Single Cell Multiome ATAC + Gene Expression Kit done on bone marrow mononuclear cells, which contains true label. We have run our method on this data and compared with Seurat. The result are displayed in the supplementary figure S10.*

"We also tested our method on the bone marrow mononuclear cells (BMMC) data from NeurIPS competition [39], which include 22,463 cells. Supplementary Figure S10 shows the clustering and 2D embedding results and comparison with Seurat. The clustering of scREG is more consistent with the ground truth label and achieve 0.7649 in NMI and 0.6549 in ARI, which are higher than Seurat (NMI = 0.7409, ARI = 0.5419). Overall, scREG identifies cell populations with high accuracy on different datasets."

Comment 4. The authors report performances of scREG_bl the version of scREG without cis-regulatory term. To my understanding, scREG_bl already exist as a method and it is called intNMF (<https://journals.plos.org/plosone/article?id=10.1371/journal.pone.0176278>). The authors should cite this method and call it with its name in their comparison.

Response: *We agree reviewer that the scREG_baseline method has the same objective function as intNMF. We have added this description in this submission and replaced the method name by intNMF. The intNMF method is one of the earliest methods for single cell multi omics data integration and has great performance on all the tested datasets. We have replaced the name scREG_baseline as intNMF in Figure 2, Figure 3, and supplementary Figure S1.*

“To test the effectiveness of the cis-regulatory potential term, we build a baseline model. We use the same strategy as described above to choose the tuning parameter λ_1 . This model has the same objective function as intNMF [36]. Thus, we call it this baseline method as intNMF.”

Comment 5. The authors should provide in the methods all the details concerning the procedure followed in the paper. Namely, how they evaluated the quality of the clusters, with which scores and applied to what etc.

Response: Thanks for your comments. We have added some detail of the model parameters and the evaluation metrics in the Method section.

“Model parameters

We have several parameters for scREG package, and all of them have default values. For all analysis in this manuscript, we use the default parameters. The number of RE-TG pairs used for the construction of the R matrix has a default value of 10000. The dimension of the H matrix has default value of 100. The hyper parameters α and β have default value of 1. Parameter C, the number of nearest neighbors used in the clustering analysis, has default value of square root of number of cells. For scAI method, we use following parameters “K = 100, nrun = 1, do.fast = T”. Here we use 100 dimensions to make it consistent with the other methods. For MOFA+ analysis, we followed the tutorial of MOFA website https://raw.githubusercontent.com/bioFAM/MOFA2_tutorials/master/R_tutorials/10x_scRNA_scATAC.html. Only difference is that we set K=100 to make it consistent with other methods. For Seurat we follow the tutorial of Seurat website https://satijalab.org/seurat/articles/weighted_nearest_neighbor_analysis.html.

Evaluation metrics

To evaluate the dimension reduction, we use silhouette index[50] for 14 -dimensions matrix for PBMC data. For PCA and LSA, we directly reduce the dimension to 14. For scAI, MOFA, and scREG, we reduce dimension to 100 and further reduce it to 14 by PCA. For clustering evaluation, we use NMI[51] and ARI[52] to compare the clustering label and ground truth. For internal clustering evaluation, we use Callinski-Harabasz index[53], Davies-Bouldin index[54], Silhouette index, and Modularity Q[55]. The inputs are clustering label and cell-cell distance matrix (calculated on the 2D embedding or top 20 PCs) on gene expression or chromatin accessibility. “

Comment 6. The authors talk in the paper about gene regulatory networks, but what they are really computing are cis-regulatory networks. The difference in between the two should be clarified and gene regulatory networks should not be used in the paper.

Response: We apologize the for misunderstanding. In this submission, the “gene regulatory networks” is not used in the paper.

Minor points:

Comment 1. In response to my comment 1, the authors affirm that MOFA+ is a NMF-based method. This is not true. MOFA is based on Bayesian Factor Analysis.

Response: We apologize for our mistake. MOFA is Bayesian Factor Analysis base method rather than NMF.

Comment 2. Relative to my comment 4, the authors should discuss the limitations of their choice. Namely, using labels based on scRNA-seq data will associate better performances to methods that are more driven by scRNAseq in the integration. The labels provided by 10X are thus not perfect labels for the evaluation of integrative methods, even if I agree that they are the only solution currently available.

Response: We agree reviewer that the ground truth used for our analysis are not perfect labels for evaluation. We have discussed this limitation in this submission.

"Last, we discuss the limitation of our analysis. To evaluate scREG method, we use the manually annotated cell labels. There is no guarantee that this label is 100% correct so that it may affect the results."

Third round of review

Reviewer 2

I thank the authors for the additional tests included in the paper. I still find the paper quite confusing, mixing clustering based on single omics with integrative clustering in all sections. I would prefer a more clear organization of the paper. In addition:

Regarding comment 1: Also in section “scREG identifies the cell populations with high accuracy” when ground-truth labels are available, average SI could be added to ARI and NMI for performance evaluation. In this case the SI would evaluate if cells belonging to the same population (based on external annotation) are closer than cells from different populations. So the quality of the dimensionality reduction provided by different methods independently of the clustering performances.

Regarding comment 2: Regarding label-free evaluation of the clusters I do not agree with the statement of the reviewers “If we calculate this distance matrix based on the reduced dimension matrix from scREG, the result reflects the effectiveness of the Louvain clustering method rather than the scREG method. We agree that the distance calculation on Umap may not reflect the real distance. In this revised version, we also calculate distance based on top 20 principal components. The results are shown in Supplementary Figure S9 and which is mostly consistent with the previous Umap-based result.”.

Working on the reduced dimension matrix from scREG would be more appropriate to this reviewer. First, it is not true that working with the reduced dimension matrix from scREG would correspond to evaluate the quality of Louvain. Indeed the performances of the Louvain clustering will be determined by the quality of the cell-to-cell distance that the various methods infer (corresponding to the reduced dimension matrix from scREG, factor matrix from MOFA etc). Testing the quality of the method on UMAP or PCA output is to me completely incorrect. As it would alter the cell-to-cell distances inferred by the various methods and would thus evaluate more the quality of UMAP/PCA rather than of the designed method. In addition, normally works evaluating multi-omics integration through matrix factorisation check the quality of the reduced dimension matrix as can be seen in:

<https://www.embopress.org/doi/full/10.15252/msb.20178124>

<https://www.nature.com/articles/s41467-020-20430-7>

In addition, the clustering results presented in section “scREG identifies the cell populations with high accuracy” are highly affected by the resolution parameter to be provided to Louvain clustering. Which resolution has been chosen in Louvain for all methods? Is this the default value? Results of NMI and ARI for various resolution levels would be more appropriate, as Seurat and MOFA+ could get results equal or better than scREG for some resolution values.

Regarding comment 3: which ground-truth the authors used? The data from NeurIPs challenge are provided with two ground-truths. A classification based on ATAC and a classification based on RNA. This information should be provided.

Reviewer #2:

I thank the authors for the additional tests included in the paper. I still find the paper quite confusing, mixing clustering based on single omics with integrative clustering in all sections. I would prefer a more clear organization of the paper.

Comment 1: Also in section “scREG identifies the cell populations with high accuracy” when ground-truth labels are available, average SI could be added to ARI and NMI for performance evaluation. In this case the SI would evaluate if cells belonging to the same population (based on external annotation) are closer than cells from different populations. So the quality of the dimensionality reduction provided by different methods independently of the clustering performances.

Response: We agree with you on the point that dimension reduction should be evaluated independently of clustering. We are evaluating the dimension reduction and clustering in two separate sections. The average SIs for different methods had already been discussed in the section “scREG performs cross-modalities dimension reduction by data integration” based on your comments in the second review. In the section “scREG identifies the cell populations with high accuracy”, we are discussing the performance of the cell clustering aspect of our method. Please see the following statement and Supplementary Figure S1.

“Figure S1 shows the comparison of average SI on all the cell types and an average SI of different dimension reduction methods. Except for the effector CD8 cell, on which all methods generally have a poor performance, our method scREG performs better on most of the cell types, SI range from 0.2371 to 0.9758, with a highest overall average (0.5614).”

Comment 2.1: Regarding label-free evaluation of the clusters I do not agree with the statement of the reviewers “If we calculate this distance matrix based on the reduced dimension matrix from scREG, the result reflects the effectiveness of the Louvain clustering method rather than the scREG method. We agree that the distance calculation on Umap may not reflect the real distance. In this revised version, we also calculate distance based on top 20 principal components. The results are shown in Supplementary Figure S9 and which is mostly consistent with the previous Umap-based result.”. Working on the reduced dimension matrix from scREG would be more appropriate to this reviewer. First, it is not true that working with the reduced dimension matrix from scREG would correspond to evaluate the quality of Louvain. Indeed the performances of the Louvain clustering will be determined by the quality of the cell-to-cell distance that the various methods infer (corresponding to the reduced dimension matrix from scREG, factor matrix from MOFA etc). *Testing the quality of the method on UMAP or PCA output is to me completely incorrect. As it would alter the cell-to-cell distances inferred by the various methods and would thus evaluate more the quality of UMAP/PCA rather than of the designed method.* In addition, normally works evaluating multi-omics integration through matrix factorisation check the quality of the reduced dimension matrix as can be seen in: <https://www.embopress.org/doi/full/10.15252/msb.20178124> <https://www.nature.com/articles/s41467-020-20430-7>.

Response: We agree that the performances of the Louvain clustering will be determined by the quality of the cell-to-cell distance that the various methods infer. If the cell label ground truth is available, one can evaluate the cell-to-cell distance provided by different methods by comparing the clustering labels with the ground truth. We have performed this comparison in PBMC data from 10X Genomics and the BMMC data from the NeurIPS competition. However, such validation becomes complicated when the

ground truth label is not available. If a ground truth label is not available, we cannot directly evaluate the performance of clustering. Thus, we cannot evaluate the quality of the reduced dimension matrix. The internal clustering evaluation metrics have been used for evaluating the clustering accuracy in many papers. However, these metrics cannot be used for the evaluation of the reduced dimension matrix since they are calculating the consistency of the reduced dimension matrix and the clustering labels. This is an underappreciated statistical issue so we generated two artificial examples to make it clear (Please see the Supplementary Note for more detail).

The internal clustering evaluation metrics (i.e. Silhouette Index and Davies–Bouldin index) evaluate the clustering results based on the cell-to-cell distance. Those indexes are comparing the within-cluster distance and the between-cluster distance. If we understand the reviewer’s comment correctly, the reviewer wants us to compare different embeddings (i.e. reduced dimension matrix from scREG or MOFA) using the internal clustering evaluation metrics by taking the embedding and corresponding clustering results as input. We didn’t conduct this comparison since **those internal clustering evaluation metrics are designed for evaluating different clustering results based on the same cell-to-cell distance matrix rather than different cell-to-cell distance matrices**. When the distance matrices are different, the comparison becomes meaningless. To illustrate this, we generate the following artificial example.

We simulated a dataset consisting of 400 cells represented by 20 features (i.e. correspond to PCs in real data). There are 4 cell types and each has 100 cells. We generate four different 20-dimension embeddings and perform clustering analysis based on these different embeddings. Here is the characteristic of 4 embeddings (Please see the detail of data generation in the Appendix of this letter).

Embedding 1: Cell types 1 and 2 are separated clearly but cell types 3 and 4 are not separated. Simulating top 20 PCs of the scRNA-seq data.

Embedding 2: Cell types 1 and 2 are not separatable, but cell types 3 and 4 are separated clearly. Simulating top 20 PCs of the scATAC-seq data.

Embedding 3: All four cell types are separated. Simulating a good joint dimension reduction method.

Embedding 4: All four cell types could be detected but have bigger noise. Simulating a bad joint dimension reduction method.

In this example, some cell types are not separated in RNA-seq space, and some are not separated in ATAC-seq space. The following Figure shows the tSNE on the 4 embeddings colored by the true label. We calculate NMI and ARI to evaluate the clusterings.

Figure R1. The tSNE plot of 4 different embeddings. Color represents the cell label.

From both the figure and the accuracy measurements, we can see that embedding 3 performs the best. Then we calculate the internal clustering evaluation metrics (Calinski-Harabasz index, Davies-Bouldin index, silhouette index, and Q_modularity) based on the embeddings and clusterings for each data set. Based on the internal evaluation metrics, embeddings 3 performs the worst (DB index is the lower the better and the other 3 indexes are the higher the better). The results are not consistent with the evaluation based on the ground truth.

	CH	DB	SI	Q
Embedding1	483.8709	0.7930	0.7428	0.1639
Embedding2	204.8519	1.2749	0.5127	0.1421
Embedding3	121.0004	1.5276	0.4163	0.0973
Embedding4	351.9862	0.9426	0.6429	0.1562

From this example, we can see it is not proper to compare the different clustering results using internal clustering evaluation metrics based on different embedding data. To compare clustering with internal clustering evaluation metrics, different clustering results should be evaluated on the same embedding space (same distance matrix). Since clusterings cannot be evaluated by their own embedding space, then the next question becomes which embedding space can provide a fair comparison. Our original design is to use RNA-seq alone embedding and ATAC-seq alone embedding to evaluate the clusterings. If one clustering (say clustering based on embedding 3) is better than the other one (say clustering based on embedding 4) on **both** RNA-seq embedding (say embedding 1) based evaluation and ATAC-seq embedding (say embedding 2) based evaluation, we can say that clustering based on embedding 3 is better than clustering based on embedding 4.

Here we again use the toy example we created above to show this. Here, embedding 1 and 2 can be seen as two different modalities that capture different aspects of the variance of the data (like scRNA-seq and scATAC-seq), embedding 3 and 4 can be seen as two joint dimension reduction methods that capture both modality's information. We compare clustering 3 and 4, from embedding 3 and 4 respectively, by calculating internal clustering evaluation metrics based on embedding 1 and 2 separately. The following table shows that clustering 3 performs better than clustering 4 on all 4 evaluation metrics under both embeddings. This example shows that our original validation is meaningful in this case.

CK	Clustering 3	Clustering 4	SI	Clustering 3	Clustering 4
Embedding1	312.3238	208.7874	Embedding1	0.3912	0.2876
Embedding2	124.6341	69.1626	Embedding2	0.2646	0.1469
DB	Clustering 3	Clustering 4	Q	Clustering 3	Clustering 4
Embedding1	3.1818	3.7658	Embedding1	0.1136	0.1093
Embedding2	4.4089	5.5047	Embedding2	0.0964	0.0881

In this revision, we have removed all results based on the Umap evaluation. This artificial example of internal evolution together with a simpler example has been added as a supplementary note for explanation.

Comment2.2: In addition, the clustering results presented in section “scREG identifies the cell populations with high accuracy” are highly affected by the resolution parameter to be provided to Louvain clustering. Which resolution has been chosen in Louvain for all methods? Is this the default value? Results of NMI and ARI for various resolution levels would be more appropriate, as Seurat and MOFA+ could get results equal or better than scREG for some resolution values.

Response: Thank you for your comments. The results are shown in the Figure and the main text are clustering results under the default resolution setting, which is 0.8 for Seurat and 1 for all other methods. We added a supplementary table (Supplementary Figure S3) to compare different clustering methods under different resolution parameters ranging from 0.2 to 2.0. Indeed, the resolution parameter affects the clustering results, especially for Seurat. The scREG performs very robust under different resolution parameters and achieve the highest performance among all the method we compared.

Methods	intNMF		MOFA		scAl		Seurat		scREG	
	nmi	Ari	nmi	Ari	nmi	Ari	nmi	Ari	nmi	Ari
Resolution=0.2	0.7023	0.4158	0.7597	0.5466	0.7321	0.4467	0.8350	0.7694	0.7766	0.5886
Resolution=0.4	0.7901	0.6142	0.8039	0.7171	0.7627	0.5811	0.8303	0.7662	0.8337	0.7483
Resolution=0.6	0.8097	0.7141	0.7982	0.7114	0.7615	0.5803	0.7753	0.6351	0.8436	0.7765
Resolution=0.8	0.8035	0.7050	0.7977	0.7168	0.7845	0.6993	0.7607	0.6229	0.8417	0.7754
Resolution=1.0	0.7928	0.7056	0.7987	0.7239	0.7867	0.7044	0.7565	0.6206	0.8467	0.7881
Resolution=1.2	0.7868	0.6925	0.7634	0.6453	0.7879	0.7071	0.7551	0.6178	0.8286	0.7413
Resolution=1.4	0.7878	0.6944	0.7430	0.5377	0.7882	0.7076	0.7395	0.5613	0.8261	0.7384
Resolution=1.6	0.7903	0.6901	0.7477	0.5394	0.7860	0.7299	0.7410	0.5645	0.8186	0.7341
Resolution=1.8	0.7655	0.6435	0.7446	0.5332	0.7875	0.7325	0.7378	0.5658	0.8247	0.7345
Resolution=2.0	0.7638	0.6355	0.7451	0.5360	0.7891	0.7361	0.7360	0.5563	0.7932	0.6761

The following statement has been added to the main text:

“It is worth to notice that all the clustering methods are compared under their default resolution parameter setting, which is 0.8 for Seurat and 1 for all other methods. As clustering accuracy is highly affected by the resolution provided to Louvain clustering, we also compared the clustering performance of these five methods under different resolutions ranging from 0.2 to 2.0 (Supplementary Figure S3). The scREG performs very robust under different resolution parameters and achieves the highest performance among all the methods we compared.”

Comment 3: which ground-truth the authors used? The data from NeurIPs challenge are provided with two ground-truths. A classification based on ATAC and a classification based on RNA. This information should be provided.

Response: In our paper, we used the training data of task 3 “joint embedding”. For ground truth, we use the harmonized cell-type annotation. More detail can be found in their paper (Luecken et al, 2021). They did annotate the cell type separately on each modality to avoid relying on the joint representation method

for annotation. But in the final cell annotation, cell labels between joint modalities were harmonized. Here is the relevant description in Luecken et al, 2021.

“Following modality- and batch-specific data analysis, we harmonized the cell type annotation per batch by taking the outer product of the cluster annotation to ensure substructure present in only one modality was still preserved in the final annotations. Where cluster substructure did not agree and did not lead to a clean sub clustering, we manually evaluated which modality marker features more clearly described the specific cellular subpopulation.”

*Luecken, M. D., Burkhardt, D. B., Cannoodt, R., Lance, C., Agrawal, A., Aliee, H., ... & Bloom, J. M. (2021, August). A sandbox for prediction and integration of dna, rna, and proteins in single cells. In *Thirty-fifth Conference on Neural Information Processing Systems Datasets and Benchmarks Track (Round 2)*.

Reviewer 1:

Comment: From what the authors described in their response “In this revised version, we also calculate distance based on top 20 principal components” and in the manuscript “ As ground truth labels are not available for these data, we use internal clustering evaluation on the distance among cells in the 2D embedding and top 20 principal components of scRNA-seq and scATAC-seq to compare the scREG with the Seurat.” it seems that when 20 principal components are use, the proposed scREG is not used, then the metric is not evaluating the performance of scREG. In this case I agree with Reviewer2 that it’s more appropriate to use scREG to obtain the low-dimensional representations to calculate the distance. However it could also be the case that scREG is applied at some point, for example, to perform initial dimensionality reduction before applying PCA. But this is not clearly stated, and the authors should clarify the role of scREG in the case of using 20 principle components to calculate the distance.

Response: Thank you for your reviewing our manuscript again. The internal clustering evaluation metrics are evaluating the consistency between a clustering label and a cell-to-cell distance matrix. Here we are using the two cell-to-cell distance matrices (one from RNA-seq space and one from ATAC-seq space) to evaluate the clustering results from scREG. The clustering result is based on the reduced dimension matrix from scREG. The results show that the clustering from scREG is more consistent with the cell-to-cell distance in both scRNA-seq space and the scATAC-seq space than the Seurat clustering. The reason we didn’t use the scREG embedding to evaluate the scREG clustering is that the metrics under different embeddings are not comparable. For more detail, please see our response to reviewer 2’s comment 2. We have built an artificial example to show this comparison is not proper.

Appendix

Data generation of the artificial example:

A. Cell labels

Cell 1-100 is cell type 1, cell 101-200 is cell type 2, cell 201-300 is cell type 3 and cell 301-400 is cell type 4.

B. Top 20 PCs (embeddings).

Embedding 1:

- 1) generate three cluster centers c_1 , c_2 , and c . They are 20-dimension vectors each independently generated from the standard normal distribution.
- 2) The center of cell type 3 and cell type 4 are generated as $c_3=c+r_3$ and $c_4=c+r_4$, where r_3 and r_4 are generated from $N(0,0.1)$.
- 3) embedding of cell type i is generated as c_i+x_i , where x_i is $20*100$ size of matrices generated from $N(0,0.5)$.

Embedding 2:

- 1) generate three cluster centers c , c_3 , and c_4 . They are 20-dimension vectors each independently generated from the standard normal distribution.
- 2) The center of cell type 1 and cell type 2 are generated as $c_1=c+r_1$ and $c_2=c+r_2$, where r_1 and r_2 are generated from $N(0,0.1)$.
- 3) embedding of cell type i is generated as c_i+x_i , where x_i is $20*100$ size of matrices generated from $N(0,0.8)$.

Embedding 3:

- 1) generate three cluster centers c_1 , c_2 , c_3 , and c_4 . They are 20-dimension vectors each independently generated from the standard normal distribution.
- 2) embedding of cell type i is generated as c_i+x_i , where x_i is $20*100$ size of matrices generated from $N(0,1)$.

Embedding 4:

- 1) generate three cluster centers c_1 , c_2 , c_3 , and c_4 . They are 20-dimension vectors each independently generated from the standard normal distribution.
- 2) real embedding of cell type i is generated as c_i+x_i , where x_i is $20*100$ size of matrices generated from $N(0,1.5)$.
- 3) KNN impute the real embedding: use nearest 20 cells' average real embedding to generate the new embedding. This will generate more structured embedding data.

C. Clusterings

We use Euclidean distance to calculate the distance between cells first and then calculate Jaccard distance based on their shared nearest neighbors ($k=20$). We cluster the cells using the Louvain algorithm under the default settings.